# Semi-Supervised Blind Quality Assessment with Confidence-quantifiable Pseudo-label Learning for Authentic Images

Yan Zhong [1 2]  Chenxi Yang [1 2]  Suyuan Zhao [3]  Tingting Jiang [2 4] ✉

## Abstract

This paper presents CPL-IQA, a novel semi-supervised blind image quality assessment (BIQA) framework for authentic distortion scenarios. To address the challenge of limited labeled data in IQA area, our approach leverages confidence-quantifiable pseudo-label learning to effectively utilize unlabeled authentically distorted images. The framework operates through a preprocessing stage and two training phases: first converting MOS labels to vector labels via entropy minimization, followed by an iterative process that alternates between model training and label optimization. The key innovations of CPL-IQA include a manifold assumption-based label optimization strategy and a confidence learning method for pseudo-labels, which enhance reliability and mitigate outlier effects. Experimental results demonstrate the framework's superior performance on real-world distorted image datasets, offering a more standardized semi-supervised learning paradigm without requiring additional supervision or network complexity.

## 1. Introduction

Recent years have witnessed the proliferation of real-world image datasets, which is the cornerstone of computer vision research and application (Yue et al., 2022). However, there may occur distortions during the process of image acquisition, transmission, and processing (Prabhakaran & Swamy, 2023), leading to degradation in the visual quality

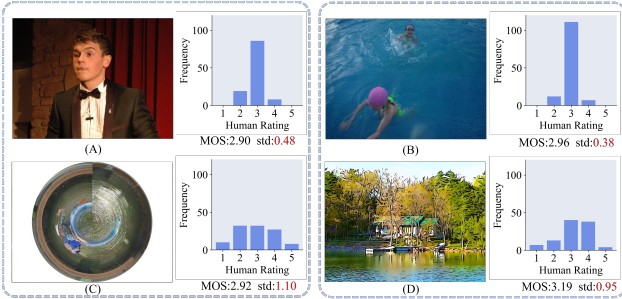

Figure 1. Real-world images with similar MOS values may differ greatly in score distribution and standard deviation (std), such as images (A) and (C), as well as images (B) and (D). These four images are all taken from the authentically distorted image database KonIQ-10k (Hosu et al., 2020).

of the processed image. Therefore, plentiful image quality assessment (IQA) methods have emerged. According to the amount of available reference images, existing objective IQA methods can be categorized as Full-Reference approaches(FR-IQA), Reduced-Reference approaches(RR-IQA) and No-Reference approaches (NR-IQA) (Zhai & Min, 2020). Since it is difficult to obtain the undistorted version of authentically distorted images in practical applications, growing attention has been given to NR-IQA (also termed as Blind IQA, i.e.BIQA) (Su et al., 2020), which can be applied in various domains, such as image denoising (Tian et al., 2020), restoration (Li et al., 2023) and generation (Elasri et al., 2022). In this work, we focus on BIQA methods for authentically distorted images. Traditional BIQA methods aim to predict the image quality scores (i.e. Mean Opinion Scores, MOS) by manually extracting features from distorted images (Liu et al., 2020; Zhou et al., 2017; Jiang et al., 2017). While effective for synthesized distortions, they struggle with authentic distortions. Therefore, there have sprung up numerous studies about Deep Learning (DL) -based BIQA methods for authentically distorted images, due to its strong ability in learning and the success of fusing discriminative features in visual domains (Fang et al., 2020; Su et al., 2020; Talebi & Milanfar, 2018; Ying et al., 2020). However, the training process of these methods requires a large amount of data to prevent overfitting, while there exists no large database of authentically distorted images nowadays since the annotation process is expensive and

---
[1]School of Mathematical Sciences, Peking University, Beijing, China [2]State Key Laboratory of Multimedia Information Processing, School of Computer Science, Peking University, Beijing, China [3]Department of Computer Science and Technology, Tsinghua University, Beijing, China [4]National Biomedical Imaging Center, Peking University, Beijing, China. Correspondence to: Yan Zhong <zhongyan@stu.pku.edu.cn>, Tingting Jiang <ttjiang@pku.edu.cn>.

*Proceedings of the 42nd International Conference on Machine Learning*, Vancouver, Canada. PMLR 267, 2025. Copyright 2025 by the author(s).

time-consuming (Yue et al., 2022; Prabhakaran & Swamy, 2023; Zhang et al., 2022c).

To solve this problem in authentically distorted scenarios, some researchers try to explore unsupervised or semi-supervised BIQA methods (Prabhakaran & Swamy, 2023; Yue et al., 2022; Saha et al., 2023), which can utilize unlabeled images to boost the quality prediction performance. For unsupervised strategy, the intuitive idea is to train the BIQA model on large synthetically distorted image databases first based on Contrastive Learning (Madhusudana et al., 2022; Saha et al., 2023), then the pre-trained model is fine-tuned on specific authentically distorted image database (Prabhakaran & Swamy, 2023). Although this approach performs well in synthetically distorted scenarios, the performances on authentic images are not satisfactory. For semi-supervised strategy, some works try to train BIQA networks using both labeled and unlabeled samples based on knowledge distillation (Yue et al., 2022), which requires the score distribution of each image. However, only MOS labels are available in most authentically distorted image datasets (such as SPAQ (Fang et al., 2020)), while images with similar MOS values may correspond to a variety of score distributions (as shown in Figure 1). And the latest semi-supervised BIQA methods SSLIQA (Yue et al., 2022) and SS-IQA (Pan et al., 2024) both require additional network branches and extra data during training, leading to higher training costs. In general, the applicability of existing DL-based semi-supervised BIQA methods is limited due to the rigorous training requirements and conditions.

Therefore, we propose a novel semi-supervised BIQA framework named CPL-IQA[1] based on label propagation (LP), which can be effectively trained end-to-end only on a single branch network without extra inputs. The idea of LP is to first construct a nearest neighbor graph in the feature space based on the manifold hypothesis (Zhou et al., 2003), then the pseudo-labels of unlabeled samples are obtained by the neighbor graph and labels of limited labeled samples. However, there exists a severe obstacle to applying LP effectively in the BIQA field: The approach of label propagation is skilled in handling multi-label (vector-label) data (Zhong et al., 2021), whereas MOS label in IQA datasets is a scalar. Therefore, to train BIQA model using labeled and unlabeled data simultaneously based on LP, CPL-IQA is required to achieve two goals: (1) Reasonably convert the scalar MOS labels of labeled samples into vector labels to meet the requirement of LP. (2) Effectively predict the pseudo-labels of unlabeled images and corresponding confidence levels, which will be fed into the BIQA model for training.

For Goal (1), according to NIMA (Talebi & Milanfar, 2018), training BIQA models with the MOS distribution of each image outperforms training with the scalar label of MOS.

This fact motivates us to convert MOS label to the vector label that can reasonably simulate the MOS distribution for each sample. To achieve this, we propose a preprocessing procedure named Label Conversion in CPL-IQA, which is implemented based on *entropy minimization*.

For Goal (2), firstly, we predict the pseudo-labels of unlabeled images following the process of label propagation. Then, in order to eliminate the effects of some inaccurate pseudo-labels, we propose an *entropy*-based confidence learning method to distinguish the quality of these predicted pseudo-labels, which can effectively improve the performance of CPL-IQA. It is worth noting that no auxiliary datasets and multiple network branches are required during the training process of CPL-IQA. The main contributions are summarized as follows:

- In this paper, we propose a novel and high-applicability semi-supervised BIQA method named CPL-IQA for authentic images according to learning the proper pseudo-labels for unlabeled images based on label propagation.

- We propose an effective strategy of Label Conversion to address the obstacle to applying label propagation effectively in the BIQA field, which is implemented based on the technique of *entropy minimization*.

- We propose an effective confidence learning method, which can eliminate outliers of pseudo-labels and enhance the generalization ability of CPL-IQA.

- Extensive experiments on authentically distorted image databases are conducted to validate the applicability and effectiveness of the proposed method.

## 2. Related Works

### 2.1. Blind Image Quality Assessment (BIQA)

Traditional BIQA methods focused on manually constructing statistical features and predicting MOS labels of images by linear mapping (Liu et al., 2018; Wang et al., 2002; Saad et al., 2012; Moorthy & Bovik, 2010). With the development and wide applications of deep learning (DL), DL-based BIQA methods have become the mainstream direction of BIQA research, while these BIQA methods are "data hungry". Therefore, some unsupervised BIQA methods have sprung up (Madhusudana et al., 2022; Prabhakaran & Swamy, 2023; Saha et al., 2023; Zhao et al., 2023), which are designed based on Contrastive Learning (Le-Khac et al., 2020), and more complex BIQA network structures are proposed to improve the generalization of BIQA (Liu et al., 2017; Lin et al., 2020; Ma et al., 2017b; Zhou et al., 2022). However, these methods are trained on synthetically distorted databases, leading to limited performance on images with authentic distortions. More recently, some

---

[1]CPL is short of **c**onfidence-quantifiable **p**seudo-label **l**earning.

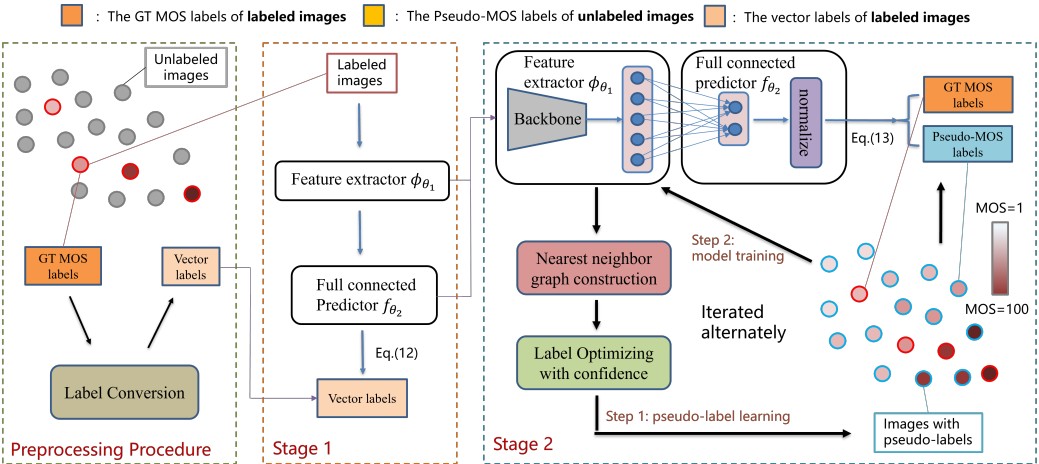

*Figure 2.* Overview of CPL-IQA. The overall training process is described in Section 3.2.

works (Zhang et al., 2024; Zhong et al., 2024a; Wang et al., 2023; Yang et al., 2024; Zhang et al., 2022a;b) try to improve the generalization of BIQA models by combining BIQA networks with specific learning paradigm, such as Continual Learning, Causal Learning, Curriculum Learning, Adversarial Learning and Semi-supervised Learning. However, these methods require much higher training costs, including the latest semi-supervised BIQA methods SSLIQA (Yue et al., 2022) and SS-IQA (Pan et al., 2024), which both require training with multiple network branches and additional datasets. Therefore, the motivation of this paper is to design a semi-supervised BIQA method that can be effectively trained end-to-end only on a single branch network without extra inputs in authentically distorted scenarios. See Appendix B.1 for more reviews.

## 2.2. Deep Semi-supervised Learning (D-SSL)

By designing different unsupervised losses for training with unlabeled samples, existing D-SSL methods can effectively deal with the scenarios of data scarcity in practical applications. One type of representative D-SSL method is known as Transductive Semi-Supervised Learning (TSSL) (Wang et al., 2016; Shi et al., 2018). Another type is designed based on label propagation (Iscen et al., 2019; Douze et al., 2018; Haeusser et al., 2017). More recently, some active learning-based semi-supervised methods have been proposed (Fan et al., 2024). Regrettably, these approaches are only suitable for classification problems and cannot be implemented in regression tasks such as IQA. Therefore, some BIQA works (Prabhakaran & Swamy, 2023; Yue et al., 2022) try to construct the unsupervised contrastive loss that can be trained with unlabeled samples, which inevitably require additional training costs with lower applicability. Different from these unsupervised methods, our proposed BIQA method can achieve both high applicability and low complexity with a more standard semi-supervised paradigm. For a detailed literature review, please refer to Appendix B.2.

## 3. Methods

We first outline the preliminaries and the overview of CPL-IQA, followed by detailed introductions of each module.

### 3.1. Preliminaries

We assume $X := (x_1, \ldots, x_l, x_{l+1}, \ldots, x_n)$ is the authentically distorted image dataset in the semi-supervised BIQA domain, where the first $l$ images are labeled and the remaining $n - l$ are unlabeled images. $Y_L := (y_1, \ldots, y_l)$ are the MOS labels of the first $l$ samples. The goal of our semi-supervised BIQA method is to train an effective MOS prediction model with unlabeled images $X_U := (x_{l+1}, \ldots, x_n)$, labeled images $X_L := (x_1, \ldots, x_l)$ and their labels $Y_L$.

### 3.2. Overview of CPL-IQA

As shown in Figure 2, the basic idea of our approach is learning the confidence-quantifiable pseudo-labels for unlabeled samples $X_U$ to deal with the insufficient label problem. The whole framework consists of a preprocessing procedure named Label Conversion and two training stages. Stage 2 includes two steps: Model Training and Label Optimizing, which are iterated alternately until convergence.

**In the Label Conversion procedure before training**, due to the superior values of score distributions of images in training, we transform MOS labels into vector representations by *entropy minimization*, which can be regarded as the simulation of score distributions. **In Stage 1**, the feature extractor $\phi_{\theta_1}$ and label predictor $f_{\theta_2}$ are trained together through labeled images $X_L, Y_L$ and Eq. 10, ResNet101 (He et al., 2016) is chosen as the backbone. **In the Label Optimizing step of Stage 2**, the features of all the samples $X$ can be obtained according to the current feature extractor $\phi_{\theta_1}$. Then a nearest neighbor graph structure is constructed in the feature space based on the manifold assumption (Zhou et al., 2003), through which the latent semantic

representations of these samples can be created to learn the proper pseudo-labels $\hat{Y}_U$ for unlabeled images. **In the Model Training step of Stage 2**, the feature extractor $\phi_{\theta_1}$ and predictor $f_{\theta_2}$ are further trained with $X_L, Y_L, X_U$ and the predicted pseudo-labels $\hat{Y}_U$. In order to improve the utilization of pseudo-labels, we allot a confidence weight for each pseudo-label during this training step in Stage 2.

### 3.3. Modules

In this section, we will introduce the details of each module about our proposed CPL-IQA method, including modules of Label Conversion, Nearest Neighbor Graph Construction, Label Optimizing, Confidence of Labels, Loss Functions, Alternate Iterative Training, Analysis and Discussion about the proposed method CPL-IQA.

#### 3.3.1. LABEL CONVERSION.

The motivations of this module lie in the two main aspects: (1) The core idea of our method is to learn the pseudo-label and corresponding confidence level for each unlabeled training sample based on label propagation, which is particularly adept at handling multi-label or vector-label data. (2) According to NIMA (Talebi & Milanfar, 2018), training with the vector label representing the MOS distribution has demonstrated more advanced effectiveness compared to training with the scalar label of MOS. Therefore, we assume that the confidence levels of labels in labeled samples are high enough, and the MOS labels of labeled samples are converted into vector representations of simulated MOS distribution labels for each sample by *entropy minimization*. Although we admit that the conversion of MOS labels to vector labels of MOS distributions could be a one-to-many mapping without any prior information, *entropy minimization* is reasonable since it results in a more concentrated simulated label distribution, suggesting lower variance and higher confidence, which fits well with the above assumption of high confidence level, leading to a one-to-one mapping.

Concretely speaking, in order to press close to the artificially subjective BIQA method (Fang et al., 2020), we uniformly consider MOS labels as the average result of score sets $M = \{1, 2, 3, ..., 99, 100\}$, hence the value range of the specified MOS is the closed interval $[1, 100]$. For authentically distorted image databases with different ranges of MOS labels, we firstly normalize the MOS range to $[1, 100]$ by Max-min Normalization:

$$\boldsymbol{s}_i = 1 + \frac{z_i - \min(\boldsymbol{z})}{\max(\boldsymbol{z}) - \min(\boldsymbol{z})} \times 99, \qquad (1)$$

where $\boldsymbol{z} = [z_1, z_2, ..., z_n]$, and $\boldsymbol{s}_i$ is the normalized MOS label of the $i$-th sample. Then, the MOS value is replaced by a 100-dimensional vector $\boldsymbol{v}$, where each element can be viewed as a proportion of the corresponding score in the

score set $M$, so each element of the vector $\boldsymbol{v}$ is positive and the sum of that is 1.

Since the distribution of MOS labels is unavailable in many authentically distorted image databases, we can regard the *entropy* of $\boldsymbol{v}$ as the degree of uncertainty of vector labels. Therefore, the MOS labels of labeled samples are converted to vector labels by *entropy minimization*.

$$\boldsymbol{v}_i =: \underset{\mathbf{1}^T \boldsymbol{v}_i = 1, \boldsymbol{q}^T \boldsymbol{v}_i = s_i}{\arg\min} H(\boldsymbol{v}_i), \; H(\boldsymbol{p}) = -\sum_i p_i \log p_i, \tag{2}$$

where $\mathbf{1} = [1, ..., 1]^T \in \mathbb{R}^{100 \times 1}$, $\boldsymbol{q} = [1, ..., 100]^T \in \mathbb{R}^{100 \times 1}$. $\boldsymbol{v}_i$ is the label vector converted from the MOS label $s_i$ of the $i$-th sample, with all elements being positive. We denote all the converted labels as $V_L = [\boldsymbol{v}_1; ...; \boldsymbol{v}_l] \in \mathbb{R}^{100 \times l}$. $H(\boldsymbol{p})$ is *information entropy* (Greven et al., 2014) for an arbitrary random vector $\boldsymbol{p}$, also applying to the label vector $\boldsymbol{v}_i$. According to Eq. 2, converting the GT MOS value to a vector representation is a one-to-one problem without any prior information. For example, an MOS value $2.4$ and can uniquely be converted to vector label $[0; 0.6; 0.4; 0; ...; 0; 0]$.

#### 3.3.2. NEAREST NEIGHBOR GRAPH CONSTRUCTION.

In Stage 2 of CPL-IQA, features are extracted according to the current parameters $\theta_1$, including the parameters of the backbone and the following Fully Connected (FC) layer. The role of FC is to realize the low dimensional representation of extracted feature information and reduce the computation complexity of graph construction, inspired by low-rank learning (Hu et al., 2021). In our settings, the backbone is ResNet101 (He et al., 2016) and the subsequent FC layer maps the learned 2048-dimensional features extracted by ResNet101 into 256-dimensional features, denoted as $Q = [q_1, q_2, \ldots, q_n] \in \mathbb{R}^{d \times n}$ ($d = 256$). We use the $k$NN(Belkin & Niyogi, 2003) to construct the weight matrix $G$, so sample point $q_i$ is linked to $q_j$ if and only if $q_i$ is the $k$ nearest neighbor of $q_j$ in $k$NN graphs. $k$NN graph $G$ is defined as:

$$G_{ij} = \begin{cases} \exp\left(-\frac{\|q_i - q_j\|^2}{\sigma^2}\right), & \text{if } q_i \text{ is linked to } q_j, \\ 0 & , \text{ otherwise}, \end{cases} \tag{3}$$

where $\sigma$ is the hyper-parameter.

Note that $q_j$ is not necessarily one of the $k$ nearest neighbors of $q_i$ when $q_i$ is that of $q_j$. Therefore, for the sake of the subsequent learning, we construct the normalized symmetrical graph matrix $\tilde{G}$ in our method:

$$\hat{G} = \left(G + G^T\right)/2, \quad \tilde{G} = D^{-1/2} \hat{G} D^{-1/2}. \tag{4}$$

The $D$ in Eq. 4 is a diagonal matrix, where the element in the $i$-th row and $i$-th column is the sum of the $i$-th row of $\hat{G}$.

### 3.3.3. LABEL OPTIMIZING.

Following the format of the converted vector labels, we initialize the vector labels of the unlabeled images as the zero vectors with 100 dimensions, denoted as $V_U = [v_{l+1}; v_{l+2}; ...; v_n]$. According to the manifold hypothesis (Zhou et al., 2003), the latent semantic representations $P \in \mathbb{R}^{n \times 100}$ of all samples can be learned through the following iterative process, where $t$ denotes the number of iterations and all elements of $P(0)$ are initialized to zero.

$$P(t + 1) = \gamma \tilde{G} P(t) + (1 - \gamma)V, \qquad (5)$$

in which $V = [V_L; V_U]$ ($V_L$ is defined after Eq. 2), and $\gamma$ is a hyper-parameter between 0 and 1, which is set as 0.99 in our method. To save computational costs, we can obtain $P^*$ (i.e. the convergent $P$) via the following assertion:

*Assertion* 3.1. Define sequence $\{P(t)\}$ as Eq. 5 with $\tilde{G}$ obtained by Eq. 4, then $\{P(t)\}$ converges to $P^*$ in Eq. 6:

$$P^* = \lim_{t \to \infty} P(t) = (1 - \gamma)(I - \gamma \tilde{G})^{-1}V. \qquad (6)$$

See Appendix A.1 for the proof. Finally, we can normalize $P^*$ and obtain the predicted pseudo-labels $\hat{Y}_U$:

$$\hat{P} = P^* / \|P^*\|_2, \ W = \hat{P}_L^{-1} Y_L, \ \hat{Y}_U = \hat{P}_U W, \quad (7)$$

where $\hat{P}_L^{-1}$ represents the pseudo-inverse of $\hat{P}_L$, since it is not a square matrix. The intuition behind Eq. 7 is as follows: After computing $P$ and $\hat{P} = [\hat{P}_L; \hat{P}_U]$, we have obtained the latent semantic representations for both labeled and unlabeled samples within the same feature space. Then, labels $Y$ can be predicted using a weight matrix $W$ as $Y = \hat{P}W$. Since the ground-truth labels $Y_L$ for labeled samples are known, we can calculate $W$ from $Y_L$ and $\hat{P}_L$, and then use $W$ to predict the labels $Y_U$ for unlabeled samples. Based on Eq. 7, our label optimizing method can handle regression tasks such as IQA. While traditional label propagation is only suitable for classification tasks (Iscen et al., 2019).

### 3.3.4. CONFIDENCE OF LABELS.

According to the normalized $\hat{P}$ in Eq. 7, we can use *information entropy* $H(\cdot)$ to estimate the confidence $\eta$ of the pseudo-labels learned in the module of Label Optimizing, then the impact of pseudo-labels with high uncertainty can be reduced for model training. For the $j$-th unlabeled sample, the confidence $\eta_j$ can be defined as:

$$\eta_j = 1 - H\left(\hat{P}_j\right) / \log(m), \ (j = l + 1, ..., n), \quad (8)$$

where $m = |M|$ is the size of set $M$. Thus, $\eta_j$ ranges between 0 and 1, with higher $\eta_j$ indicating greater confidence. On the other hand, we assign high confidence to the ground truth MOS labels of labeled samples, that is:

$$\eta_i = 1, (i = 1, ..., l). \qquad (9)$$

The setting of Eq. 9 echoes the operation of Label Conversion for labeled samples.

### 3.3.5. LOSS FUNCTIONS.

As shown in Figure 2, our proposed CPL-IQA method includes two training stages.

Stage 1 of CPL-IQA is a supervised learning process trained with labeled images $X_L$ and their labels $Y_L$, the loss function of which can be formulated as:

$$L_{s1}\left(X_L, Y_L; \theta_1, \theta_2\right) := \sum_{i=1}^{l} \text{loss}_1\left(f_{\theta_2}\left[\phi_{\theta_1}(x_i)\right], EM(y_i)\right),$$
$$(10)$$

where $EM(\cdot)$ denotes the operation of Label Conversion with *entropy minimization*, $\phi$ and $f$ are feature extractor and label predictor with parameters $\theta_1$ and $\theta_2$ respectively, and $loss_1$ represents the EMD (earth mover's distance) loss (Hou et al., 2016) with $r = 2$.

Stage 2 of CPL-IQA is trained with both labeled samples $X_L, Y_L$ and unlabeled samples $X_u$ and the loss function is:

$$L_{s2}\left(X, Y_L, \hat{Y}_U; \theta_1, \theta_2\right) := \sum_{i=1}^{l} \eta_i \, \text{loss}_2\left(f_{\theta_2}\left[\phi_{\theta_1}(x_i)\right], y_i\right)$$
$$+ \sum_{j=l+1}^{n} \eta_j \, \text{loss}_2\left(f_{\theta_2}\left[\phi_{\theta_1}(x_j)\right], \hat{y}_j\right),$$
$$(11)$$

where $\hat{Y}_U := (\hat{y}_{l+1}, ..., \hat{y}_n)$ denote the learned pseudo-labels for unlabeled images $X_U$ in the Label Optimizing step of Stage 2 . And $loss_2$ represents a loss function designed based on $L_1$ loss (MAE loss).

### 3.3.6. ALTERNATE ITERATIVE TRAINING.

After Label Optimizing in Eq. 7 and confidence learning in Eq. 8, we can further train the feature extractor $\phi_{\theta_1}$ and FC predictor $f_{\theta_2}$ with Eq. 11. Since the predicted results of $f_{\theta_2}$ (with normalization) are $m$-dimensional vector labels, the learned pseudo-labels are real numbers range in interval $[1, 100]$. Therefore, $loss_2$ in Eq. 11 can be measured by:

$$\text{loss}_2\left(d_k, y_k; g\right) = L_1\left(d_k * g^T, y_k\right), (k = 1, ..., n), \ (12)$$

where $d_k = normalize\{f_{\theta_2}\left[\phi_{\theta_1}(x_k)\right]\}$ and $g = [1, 2, ..., m]$ is the vector of label levels, $m = 100$ in our method. And $L_1(\cdot)$ denotes the $L_1$ loss (MAE loss).

Therefore, the process of alternate training in Stage 2 can be iterated through the two steps: **Step 1**: Label Optimizing. According to image features extracted by the latest updated model, a new $k$NN graph is constructed by Eqs. 3-4. Then the pseudo-labels of unlabeled samples are learned by Eqs. 6-7. **Step 2**: Model Training. After calculating the confidence of pseudo-labels by Eq. 8, the current model is further trained with Eqs. 11-12.

The pseudo-code of CPL-IQA is summarized in Algorithm 1 in Appendix C, where line 1, lines 1-1. lines 1-1 and lines 1-1 are the processes of Label Conversion, Stage 1, Step 1 and Step 2 of Stage 2, respectively.

### 3.3.7. ANALYSIS AND DISCUSSION.

In the process of label propagation (Zhou et al., 2003), features are normalized to construct the manifold structure of samples (Iscen et al., 2019) by cosine similarity, which is not suitable for regression and IQA tasks, since this kind of manifold structure may wreak havoc on the distribution of regression labels (see Appendix E.3 for details). Instead of cosine similarity, we construct the nearest neighbor graph by the original extracted feature information, which is more compatible with IQA tasks. An interesting finding is that the limiting value Eq. 6 in the iterative process of Eq. 5 can be regarded as the optimal solution of a specific regularization framework, which describes the process of label propagation from the perspective of optimization. Here we have the following assertion.

*Assertion* 3.2. In the iterative process of Eq. 5, the limiting value Eq. 6 can be regarded as the optimal solution of the regularization framework Eq. 13 (with $\mu > 0$ in Eq. 14).

$$P^* = \arg\min_{P \in \mathcal{P}} \mathcal{H}(P), \tag{13}$$

$$\mathcal{H}(P) = \sum_{i,j=1}^{n} G_{ij} \left\| \frac{1}{\sqrt{D_{ii}}} P_i - \frac{1}{\sqrt{D_{jj}}} P_j \right\|^2 + \mu \sum_{i=1}^{n} \|P_i - V_i\|^2. \tag{14}$$

See Appendix A.2 for proof. The first and second terms of Eq. 14 can be regarded as the *smoothness constraint* and *fitting constraint* (Zhou et al., 2003), respectively.

## 4. Experiments

In this section, we conduct experiments on authentically distorted images to validate the superiority of CPL-IQA.

### 4.1. Experimental Settings

#### 4.1.1. DATASETS AND EVALUATION METRICS

We perform the main experiments on four representative authentically distorted image databases, including KonIQ-10K (Hosu et al., 2020), LIVE-C (Ghadiyaram & Bovik, 2015), NNID (Xiang et al., 2019) and SPAQ (Fang et al., 2020). More details of these datasets are shown in Appendix D. In addition to the main experiments, more databases in Table 9 are used in more extra experiments in Appendix E, including BID (Ciancio et al., 2010) and KADID-10K (Lin et al., 2019). We evaluate BIQA models by four typical metrics, including Pearson Linear Correlation Coefficient (PLCC), Spearman Rank-order Correlation Coeffi-

*Table 1.* Performance comparison on KonIQ-10K with the proportion 1:3:1 of samples division. The best results are highlighted in bold (same in the later tables).

| Methods | | | PLCC | SRCC |
|---|---|---|---|---|
| Traditional BIQA | Supervised | NIQE | 0.300 | 0.276 |
| | | BRISQUE | 0.581 | 0.541 |
| | | GWH-GLBP | 0.581 | 0.541 |
| | | SSEQ | 0.326 | 0.303 |
| DL-based BIQA | Supervised | CNNIQA | 0.654 | 0.635 |
| | | WaDIQaM | 0.665 | 0.644 |
| | | PAQ-2-PIQ | 0.728 | 0.718 |
| | | NSSADNN | 0.665 | 0.549 |
| | | GraphIQA | 0.862 | **0.845** |
| | | MB-CNN | 0.609 | 0.600 |
| | | MUSIQ | 0.864 | 0.838 |
| | | Causal-IQA | 0.853 | 0.837 |
| | Unsupervised | CONTRIQUE | 0.768 | 0.775 |
| | | Re-IQA | 0.782 | 0.804 |
| | Semi-Supervised | SSLIQA | 0.867 | 0.841 |
| | | SS-IQA | 0.854 | 0.829 |
| | | Semi-IQA | 0.864 | 0.839 |
| | | **Ours** | **0.873** | **0.845** |

cient (SRCC), Kendall Rank-order Correlation Coefficient (KRCC), and Root Mean Squared Error (RMSE).

#### 4.1.2. IMPLEMENTATION DETAILS

In the CPL-IQA, we choose ResNet101 as the backbone for a compelling comparison with SSL-IQA (Yue et al., 2022), which uses ResNet101 (He et al., 2016) as one branch. In Label Conversion, we take score set $M = \{1, 2, 3, ..., 99, 100\}$ and $m = 100$ is the cardinality of $M$. $k$ is set to 10 and $\sigma = 500$ in the Nearest Neighbor Graph Construction with $k$NN, and the graph $G$ in Eq. 3 is computed with the FAISS library (Johnson et al., 2019). We let $\gamma = 0.99$ in the process of Label Optimizing. The dimension $d$ of features extracted by the FC layer after the backbone is 256, which can improve computing efficiency in graph construction.

Our CPL-IQA is trained with Pytorch library on two Intel Xeon E5-2609 v4 CPUs and four NVIDIA RTX 2080Ti GPUs. The batch size $B = 64$ in Stage 1, and Stage 2 is performed with $B = B_L + B_U$, where $B_L = 8$ and $B_U = 56$ denote the number of labeled and unlabeled images in one batch. The training is conducted for just 10 epochs in total with SGD optimization, including 5 epochs in Stage 1 and 5 epochs in Stage 2. Meanwhile, we resize all the images into $256 \times 256$ and randomly crop 10 sub-images to the size of $224 \times 224$, and we initialize the backbone by the pre-training weights obtained by the classification task on ImageNet (Deng et al., 2009) before training in Stage 1.

### 4.2. Performance Comparison

#### 4.2.1. MAIN RESULTS

We divide the image set of KonIQ-10K by 1:3:1, which corresponds to the ratio of the number of training im-

*Table 2.* The test results of cross-data experiments conducted on LIVE-C and NNID. All the CNN-based methods are trained with 20% labeled images and 60% unlabeled images sampling from KonIQ-10K.

| Methods | | LIVE-C | | NNID | |
|---|---|---|---|---|---|
| | | PLCC | SRCC | PLCC | SRCC |
| Supervised | CNNIQA | 0.513 | 0.485 | 0.597 | 0.584 |
| | WaDIQaM | 0.538 | 0.535 | 0.704 | 0.702 |
| | PAQ-2-PIQ | 0.528 | 0.506 | 0.715 | 0.712 |
| | NSSADNN | 0.439 | 0.426 | 0.733 | 0.731 |
| | GraphIQA | 0.619 | 0.591 | 0.728 | 0.727 |
| | MB-CNN | 0.481 | 0.459 | 0.553 | 0.548 |
| | MUSIQ | 0.655 | 0.672 | 0.701 | 0.723 |
| | Causal-IQA | 0.742 | 0.705 | 0.748 | 0.767 |
| Unsupervised | CONTRIQUE | 0.598 | 0.612 | 0.584 | 0.679 |
| | Re-IQA | 0.612 | 0.593 | 0.624 | 0.770 |
| Semi-Supervised | SSLIQA | 0.706 | 0.695 | 0.771 | 0.770 |
| | SS-IQA | 0.731 | 0.705 | 0.743 | 0.759 |
| | Semi-IQA | 0.718 | 0.684 | 0.723 | 0.735 |
| | **Ours** | **0.777** | **0.721** | **0.772** | **0.773** |

*Table 3.* Impacts of Label Conversion, the weight of Label Confidence and Label Optimizing in Stage 2.

| Methods | Metrics | KonIQ-10K (1:3:1) | SPAQ (1:8:1) | SPAQ (2:7:1) |
|---|---|---|---|---|
| Label-only | PLCC | 0.713 | 0.812 | 0.823 |
| | SRCC | 0.719 | 0.813 | 0.818 |
| | KRCC | 0.502 | 0.589 | 0.616 |
| | RMSE | 9.979 | 13.267 | 11.665 |
| Ours (Stage 1) | PLCC | 0.850 | 0.873 | 0.888 |
| | SRCC | 0.822 | 0.869 | 0.887 |
| | KRCC | 0.619 | 0.672 | 0.686 |
| | RMSE | 7.288 | 10.284 | 9.466 |
| Ours (Full) (weights ×) | PLCC | 0.862 | 0.881 | 0.898 |
| | SRCC | 0.829 | 0.883 | 0.897 |
| | KRCC | 0.630 | 0.689 | 0.699 |
| | RMSE | 7.068 | 10.009 | 8.953 |
| Ours (Full) (weights ✓) | PLCC | **0.873** | **0.896** | **0.903** |
| | SRCC | **0.845** | **0.893** | **0.902** |
| | KRCC | **0.652** | **0.703** | **0.709** |
| | RMSE | **6.728** | **9.220** | **8.911** |

ages with labels, training images without labels and test images, respectively, to conduct the comparative experiments to analyze the effectiveness of CPL-IQA. Sixteen advanced BIQA methods are used to compare with CPL-IQA, which consist of four traditional BIQA methods (including NIQE (Mittal et al., 2012b), BRISQUE (Mittal et al., 2012a), GWH-GLBP (Li et al., 2016) and SSEQ (Liu et al., 2014)) and twelve DL-based BIQA methods (including PAQ-2-PIQ (Ying et al., 2020), CNNIQA (Kang et al., 2014), WaDIQaM (Bosse et al., 2017), NSSADNN (Yan et al., 2019), GraphIQA (Sun et al., 2023a), MB-CNN (Pan et al., 2022), CONTRIQUE (Madhusudana et al., 2022), MUSIQ (Ke et al., 2021), Causal-IQA (Zhong et al., 2024a), Re-IQA (Saha et al., 2023), SSLIQA (Yue et al., 2022), SS-IQA (Pan et al., 2024)), and Semi-IQA (Li et al., 2024). Semi-IQA, SS-IQA and Re-IQA are the SOTA semi-supervised BIQA methods, CONTRIQUE and Re-IQA are the SOTA unsupervised BIQA methods[2]. Other supervised methods are trained with the corresponding 20% labeled images for the sake of fairness. Finally, the comparison results between CPL-IQA and other methods are displayed in Table 1. We can observe that, compared with DL-based methods, the general performances of traditional methods are fairly unsatisfactory. And our CPL-IQA has achieved better results than other competing methods. Besides, CPL-IQA with only one branch performs better than SSLIQA and SS-IQA (consisting of two network branches), which illustrates the effectiveness of CPL-IQA.

### 4.2.2. CROSS-DATA EXPERIMENTS

In addition, we further perform a set of cross-data experiments to examine the generalization ability of CPL-IQA. Specifically, we directly test above mentioned CNN-based

---

[2]Unsupervised model is firstly trained on an unlabeled training set, then fine-tuned on a labeled training set from the same dataset.

models trained on KonIQ-10K (sample proportion is 1:3:1) on two unseen authentically distorted image databases: LIVE-C and NNID. To echo models trained on KonIQ-10K, the MOS labels of images in LIVE-C and NNID are normalized to 1-100 by Eq. 1 and these images are randomly cropped to the size of $224 \times 224$ before testing. And the results are displayed in Table 2, which shows that CPL-IQA can perform stably on unseen databases with one-up generalization ability. Peculiarly, the PLCC value of CPL-IQA on LIVE-C is more than 10% of the second place, which indicates its advantage in generalization for the authentically distorted IQA tasks.

### 4.3. Ablation Study

We investigate the impacts of different components of CPL-IQA. Firstly, we record the impact of each stage on the final results on KonIQ-10k and SPAQ with different split ratios, which are shown in Table 3. The Label-Only (Fang et al., 2020) method is trained by ResNet101 and $L_1$ loss only on labeled samples, and Ours (Stage 1) and Ours (Full) denote the training results of CPL-IQA in Stage 1 and Stage 2, respectively. The "weights ×" denotes the confidence of all samples set as 1 in Stage 2, and "weights ✓" means confidence is calculated by Eq. 8. Looking down from the top in Table 3, the performances are gradually improving, illustrating the effectiveness of Label Conversion, the weight of Label Confidence, and Label Optimizing in Stage 2.

In addition, we study the impact of cardinal number $m$ of the score set $M$ mentioned in Section 3.3 and the split ratio of image samples, the results of which are shown in Table 4 and Table 5. Looking from left to right, we can observe that the larger the ratio of labeled training samples, the better

*Table 4.* Impact of the cardinality $m$ of score set $M$.

|  | KonIQ-10K(1:3:1) | | | SPAQ(1:8:1) | | |
|---|---|---|---|---|---|---|
| $m$ | 10 | 20 | 100 | 10 | 20 | 100 |
| PLCC | 0.850 | 0.872 | **0.873** | 0.884 | 0.892 | **0.896** |
| SRCC | 0.821 | 0.837 | **0.841** | 0.880 | 0.888 | **0.893** |
| KRCC | 0.627 | 0.644 | **0.652** | 0.686 | 0.697 | **0.703** |
| RMSE | 7.081 | 6.788 | **6.728** | 9.673 | 9.345 | **9.220** |

*Table 5.* Impact of the split ratio of datasets (fixed $m = 20$).

| $m$=20 | KonIQ-10K | | | SPAQ | | |
|---|---|---|---|---|---|---|
| Ratio | 1:7:2 | 2:6:2 | 3:5:2 | 1:8:1 | 2:7:1 | 3:6:1 |
| PLCC | 0.845 | 0.872 | **0.875** | 0.892 | 0.902 | **0.905** |
| SRCC | 0.813 | 0.837 | **0.848** | 0.888 | 0.901 | **0.903** |
| KRCC | 0.620 | 0.644 | **0.661** | 0.697 | 0.706 | **0.716** |
| RMSE | 7.284 | 6.788 | **6.700** | 9.345 | 8.935 | **8.746** |

performance, and the same goes for cardinal number $m$.

### 4.4. Visualized Analysis

We further investigate the quality of pseudo-labels learned by Eq. 7 during each iteration in Stage 2. The whole visual analysis experiments are conducted on KonIQ-10K (1:3:1) with $m = 100$. **On the one hand**, we record the performances of pseudo-labels learned by Eq. 7 in each epoch of Stage 2, including PLCC, SRCC, KRCC and RMSE according to ground-truth (GT) labels. Meanwhile, the performances of directly predicted labels by the model in the current epoch are recorded. The above results are shown in Figure 3, from which we can observe that: (i) pseudo-labels learned by Eq. 7 are almost always more effective than that predicted by the network. (ii) the performance of pseudo-labels predicted by the network is gradually improving. Therefore, the strategy of Alternate Iterative Training and the method of Label Optimizing can be proven to be effective. **On the other hand**, we make a comparison between the distribution of pseudo-labels learned by Eq. 7 and that of ground-truth (GT) MOS labels, which are shown in Figure 4. It can be observed that the distribution of pseudo-labels is almost consistent with that of the GT , which proves the effectiveness of Label Optimizing once again. Note that achieving the consistency between the distributions of pseudo-labels and GT labels is not trivial. Traditional label propagation methods, tailored for classification perform poorly (Appendix E.3), highlighting the superior adaptability of our proposed methods to IQA tasks.

### 4.5. More Experimental Results

We also conducted additional experiments to further explore several aspects, including (1) the Results with Different Backbones, (2) the Visualization of Label Confidence, (3) the impact of Cosine Similarity-based Manifold Structure on the experimental outcomes, (4) the Impact on CPL-IQA of $k$ in Eq. 3, (5) performances of CPL-IQA

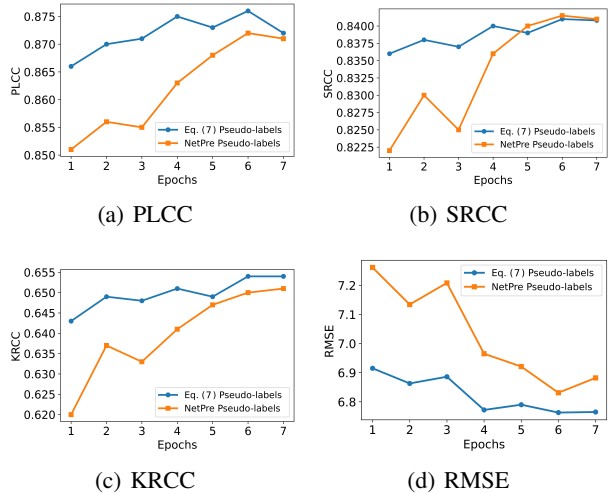

(a) PLCC  (b) SRCC

(c) KRCC  (d) RMSE

*Figure 3.* Comparison of performances between pseudo-labels learned by Eq. 7 (blue lines) and network prediction (yellow lines) according to ground-truth MOS labels on KonIQ-10K (1:3:1) in each epoch of Stage 2.

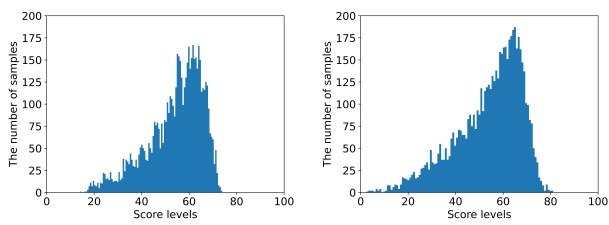

(a) Pseudo-labels via Eq. 7  (b) Ground-truth (GT) labels

*Figure 4.* The distribution contrast of pseudo-labels obtained by Eq. 7 for unlabeled training samples and the corresponding GT-labels, validating the accuracy of the predicted pseudo-labels.

trained with labeled and unlabeled samples from different datasets, and (6) Evaluating CPL-IQA with unlabeled training data from multiple sources. More details are shown in Appendixes E.1, E.2, E.3, E.4, E.5, and E.6 respectively.

## 5. Conclusions

This paper proposes a novel semi-supervised BIQA method CPL-IQA for real-world images with authentic distortions based on confidence-quantifiable pseudo-label learning, which includes a preprocessing procedure (Label Conversion) and a two-stage training process. In the preprocessing procedure of Label Conversion, MOS labels are transformed into vector labels according to *entropy minimization*, then a basic feature extractor is trained with limited labeled samples in Stage 1, based on which the proper pseudo-labels are learned for unlabeled images in the Label Optimizing step of Stage 2, which is executed alternately and iteratively with the step of Model Updating for continuous training until convergence. Extensive experiments are conducted to prove the superiority of the proposed CPL-IQA.

## Acknowledgements

This work is partially supported by Sino-German Center (M 0187) and the NSFC under contract 62088102. We also acknowledge High-Performance Computing Platform of Peking University for providing computational resources.

## Impact Statement

This paper presents work whose goal is to advance the field of Machine Learning and Image Quality Assessment. There may exist some potential societal consequences of our work, none of which must be specifically highlighted here.

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

## A. Proofs

### A.1. The Proof of Assertion 3.1

*Proof.* Without loss of generality, let $P(0) = V$, according to Eq. 5, we have:

$$P(t) = (\gamma \tilde{G})^{t-1} V + (1 - \gamma) \sum_{j=0}^{t-1} (\gamma \tilde{G})^j V. \quad (15)$$

Note that $0 < \gamma < 1$, and the eigenvalues of $\tilde{G}$ are in the interval $[-1, 1]$, then

$$\lim_{t \to \infty} (\gamma \tilde{G})^{t-1} = 0, \quad (16)$$

$$\lim_{t \to \infty} \sum_{j=0}^{t-1} (\gamma \tilde{G})^j = (I - \gamma \tilde{G})^{-1}. \quad (17)$$

Hence we can obtain:

$$P^* = \lim_{t \to \infty} P(t) = (1 - \gamma)(I - \gamma \tilde{G})^{-1} V, \quad (18)$$

which proves up Assertion 3.1. $\square$

### A.2. The Proof of Assertion 3.2

*Proof.* We set the derivative of $\mathcal{H}(P)$ w.r.t $P$ equal to zero, to obtain the optimal $P$ (denoted as $P^*$):

$$\frac{\partial \mathcal{H}}{\partial P}\bigg|_{P=P^*} = P^* - \tilde{G}P^* + \mu(P^* - V) = 0, \quad (19)$$

which is equivalent to:

$$P^* - \frac{1}{1 + \mu} \tilde{G} P^* - \frac{\mu}{1 + \mu} V = 0. \quad (20)$$

Hence we have:

$$(I - \frac{1}{1 + \mu} \tilde{G}) P^* = \frac{\mu}{1 + \mu} V, \quad (21)$$

where $I$ is the identity matrix. Note that $I - \frac{1}{1+\mu}\tilde{G}$ is a symmetric and invertible matrix, then we have:

$$P^* = \frac{\mu}{1 + \mu}(I - \frac{1}{1 + \mu} \tilde{G})^{-1} V. \quad (22)$$

Note that $\mu > 0$, so we can regard $1/(1 + \mu)$ and $\mu/(1 + \mu)$ as $\gamma$ and $1 - \gamma$ respectively, then Eq. 22 can be rewritten as:

$$P^* = (1 - \gamma)(I - \gamma \tilde{G})^{-1} V, \quad (23)$$

which is the same as Eq. 6 in the submitted version, and proves up Assertion 3.2. $\square$

# B. More detailed Related Works

## B.1. Blind Image Quality Assessment

Blind Image Quality Assessment (BIQA) has attracted wide attention in recent year since reference images are unavailable in authentically distorted image datasets (Wang et al., 2002). Traditional BIQA methods focused on manually constructing statistical features and predicting MOS labels of images by linear mapping (Wang et al., 2002; Liu et al., 2018; Saad et al., 2012; Moorthy & Bovik, 2010). With the development and wide applications of convolutional neural networks (CNNs) and deep learning (DL) (Chen et al., 2024; 2025; Zhong et al., 2025b; Ma et al., 2025; 2024b;a; Zhao et al., 2024; Zhong et al., 2024b), DL-based or CNN-based BIQA methods have become the mainstream direction of BIQA research (Zhu et al., 2024a;b; 2022), while these BIQA methods are "data hungry". Therefore, some works (Kang et al., 2014; Bosse et al., 2016; Ding et al., 2021) try to cope with the insufficient labeled data by dividing the image into several patches with the same MOS label. Although different weighting strategies can be utilized to integrate local scores, this operation still leads to the loss of global information. So, more complex networks are proposed to improve the generalization of BIQA (Liu et al., 2017; Lin et al., 2020; Ma et al., 2017b; Zhou et al., 2022). For example, RankIQA (Liu et al., 2017) proposed a Siamese Network to rank image pairs that are synthetically distorted, and MEON (Ma et al., 2017b) is pre-trained for distortion identification tasks. However, these methods are trained on synthetically distorted databases, leading to limited performance on images with authentic distortions.

To train the BIQA model with enough distorted image samples, some unsupervised BIQA methods have sprung up (Madhusudana et al., 2022; Prabhakaran & Swamy, 2023; Saha et al., 2023; Zhao et al., 2023), which are designed based on Contrastive Learning (Le-Khac et al., 2020), among which CONTRIQUE (Madhusudana et al., 2022) and (Prabhakaran & Swamy, 2023) can only be trained on the synthetically distorted databases, while Re-IQA (Saha et al., 2023) and (Zhao et al., 2023) can be trained directly on images with authentic distortions. Additionally, some of them try to make the most of existing supervisory signals, such as multi-task learning (Li et al., 2022; Ma et al., 2023), rank learning (Li et al., 2020), training with mixed-dataset (Sun et al., 2023b) and Multimode signals (Zhang et al., 2023). More recently, some works (Zhang et al., 2024; Wang & Ma, 2021; Wang et al., 2021; 2023; Yang et al., 2024; Zhang et al., 2022a;b) try to improve the generalization and robustness of BIQA models by combining BIQA networks with specific learning paradigm, such as Continual Learning, Active Learning, Curriculum Learning and Adversarial Learning. However, these methods require much higher training costs, including the latest semi-supervised

BIQA methods SSLIQA (Yue et al., 2022) and SS-IQA (Pan et al., 2024), which both require training with multiple network branches and additional datasets. Therefore, the motivation of this paper is to design a semi-supervised BIQA method that can be effectively trained end-to-end only on a single branch network without extra inputs in authentically distorted scenarios.

It is noteworthy to mention that the methodology proposed in this paper represents the current SOTA BIQA method. Although methods such as DEIQT (Qin et al., 2023), Re-IQA (Saha et al., 2023), CONTRIQUE (Madhusudana et al., 2022), and GRepQ (Srinath et al., 2024) utilize a limited labeled dataset for training, their core approach involves pre-training on a large-scale dataset followed by fine-tuning on a smaller dataset. Consequently, these methods are inherently not comparable to semi-supervised IQA approaches. Nevertheless, in the experiments detailed in the main text, we have compared our method with SOTA unsupervised IQA methods, including Re-IQA and CONTRIQUE, thereby thoroughly validating the efficacy of the proposed method. Moreover, although the work presented in (Zeng et al., 2018) also suggests transforming scalar Mean Opinion Score (MOS) labels into vector labels for the training of IQA models, it achieves this transformation into a normal distribution through a pre-defined variance. This approach not only is exclusively suitable for fully supervised training but also may lead to suboptimal performance due to the potential inaccuracy of the pre-set variance. In contrast, our proposed method, based on entropy minimization, is effectively applicable to semi-supervised IQA learning, offering a more adaptable and potentially more accurate framework for quality assessment.

## B.2. Deep Semi-supervised Learning

According to designing different unsupervised losses for training with unlabeled samples, existing Deep Semi-Supervised Learning (D-SSL) methods can effectively deal with the scenarios of data scarcity in practical applications. One type of representative D-SSL method is known as Transductive Semi-Supervised Learning (TSSL), which regards the labels of unlabeled samples as optimization variables, and iteratively updates them in the training process (Wang et al., 2016; Shi et al., 2018; Zhong et al., 2025a). Another type of D-SSL method is designed based on label propagation, which propagates labels of labeled data to nearby unlabeled data by constructing the Manifold Graph in the sample space (Iscen et al., 2019; Douze et al., 2018; Haeusser et al., 2017). More recently, some active learning-based semi-supervised methods have been proposed (Fan et al., 2024). Regrettably, these approaches are only suitable for classification problems and cannot be implemented in regression tasks such as IQA. Therefore, some BIQA works (Prabhakaran & Swamy, 2023; Yue et al., 2022) try to construct

**Algorithm 1** Training Process of CPL-IQA
___
**Input:** Parameters $\sigma$, $\gamma$;

The dimension number of score sets $m$;

Nearest neighbor parameter $k$ in $k$NN;

the vector of label levels $g = [1, 2, ..., m]$;

Unlabeled images $X_U := (x_{l+1}, x_{l+2}, \ldots, x_n)$;

Labeled images $X_L := (x_1, x_2, \ldots, x_l)$, and their MOS labels $Y_L := (y_1, y_2, \ldots, y_l)$.

**Output:** Feature extractor $\phi_{\theta_1}$, MOS label predictor $f_{\theta_2}$.

1: Initialize $\theta_1$ and $\theta_2$;
2: $V_L = EM(Y_L)$;
3: **For** $epoch \in [1, ..., T_1]$ **do**:
4:     $J_1 = \text{loss}_1(X_L, V_L; \theta_1, \theta_2)$;
5:     $\theta \leftarrow \theta - \alpha \nabla J_1 / \nabla \theta, (\theta = \{\theta_1, \theta_2\})$;
6: **End for**.
7: **Repeat:**
8:     Extract the features of all images: $Q = f_{\theta_1}(X)$;
9:     Construct nearest neighbor graph $\tilde{G}$ by Eqs. 3-4;
10:    Learn pseudo-labels $\hat{Y}_U$ by Eqs. 6-7;
11:    Compute confidence $\eta$ of pseudo-labels by Eq. 8;
12:    $J_2 = \text{loss}_2(X_L, Y_L; \theta, g) + \text{loss}_2\left(X_U, \hat{Y}_U; \theta, \eta, g\right)$;
      $\text{loss}_2$ is defined in Eq. 12.
13:    $\theta \leftarrow \theta - \alpha \nabla J_2 / \nabla \theta$;
14: **Until** convergence.

the unsupervised contrastive loss that can be trained with unlabeled samples. However, the training processes require additional supervisory information or network branches in these methods, which inevitably leads to additional training costs and lower applicability. Different from these unsupervised methods designed by Contrastive Learning, our proposed BIQA method can achieve both high applicability and low complexity with a more standard semi-supervised paradigm.

## C. Algorithm

The pseudo-code of CPL-IQA is summarized in Algorithm 1, where line 2, lines 3-6. lines 8-11 and lines 12-13 are the processes of Label Conversion, Stage 1, Step 1 and Step 2 of Stage 2, respectively.

## D. Datasets

In this paper, we perform experiments on four representative authentically distorted image databases:

- KonIQ-10K (Hosu et al., 2020). It includes 10,073 images with authentic distortions chosen from YFCC100M (Thomee et al., 2016). Eight depth feature-based content or quality metrics are used in the sampling process to ensure a wide and uniform distribution

*Table 6.* The detailed attributes of four authentically distorted image databases used in experiments.

| Databases | Number | MOS Range | *Distribution* |
|---|---|---|---|
| KonIQ-10K | 10,073 | [1,5] | Yes |
| LIVE-C | 1,162 | [0,100] | No |
| NNID | 2,240 | [0,1] | No |
| SPAQ | 11,125 | [0,100] | No |

of image content and quality in terms of brightness, color, contrast, and sharpness. And its quality is reported by MOS with the range of $[1, 5]$.

- LIVE-C (Ghadiyaram & Bovik, 2015). LIVE-C consists of 1162 authentically distorted images captured from many diverse mobile devices. Each image was assessed on a continuous quality scale by an average of 175 unique subjects, and the MOS labels range in $[0, 100]$.

- NNID (Xiang et al., 2019). NNID contains 2240 images with 448 different image contents captured by different photographic equipment in real-world scenarios. And the MOS labels of NNID range in $[0, 1]$.

- SPAQ (Fang et al., 2020). SPAQ includes 11,125 images taken by 66 mobile phones, which contain a wide range of distortions during shooting, such as sensor noise, blurring due to out-of-focus, motion blurring, over- or under-exposure, color shift, and contrast reduction. And the MOS labels range in $[0, 100]$.

The details of these datasets are shown in Table 6, where *Distribution* denotes whether the database provides the distribution information of the opinion scores for each image. In fact, there is no *Distribution* information in most authentically distorted image databases

## E. More Experimental Results

### E.1. Results of Different Backbones

In our experimental settings in Section 4, the backbone of CPL-IQA is set as ResNet101. And we make comparisons among the performances of CPL-IQA trained with different backbones, including ResNet18, ResNet50, ResNet101, and Vision Transformer-base (ViT-base).

As shown in Table 7, we can observe that these CPL-IQA models with different backbones all perform well, and the deeper the backbone (the more parameters), the better the experimental results. The results of ViT-base are just slightly better than that of ResNet101, but not significantly so, possibly due to the large number of parameters in ViT and the limited amount of training data, which led to overfitting.

*Table 7.* Comparison of the performances of CPL-IQA trained with different backbones.

| Methods | Metrics | KonIQ-10K (1:3:1) | SPAQ (1:8:1) |
|---|---|---|---|
| Ours | PLCC | 0.841 | 0.869 |
| (Full) | SRCC | 0.812 | 0.868 |
| (weights ✓) | KRCC | 0.605 | 0.673 |
| (AlexNet) | RMSE | 7.287 | 10.101 |
| Ours | PLCC | 0.852 | 0.886 |
| (Full) | SRCC | 0.810 | 0.884 |
| (weights ✓) | KRCC | 0.621 | 0.694 |
| (ResNet18) | RMSE | 7.126 | 9.892 |
| Ours | PLCC | 0.870 | 0.892 |
| (Full) | SRCC | 0.842 | 0.886 |
| (weights ✓) | KRCC | 0.649 | 0.695 |
| (ResNet50) | RMSE | 6.734 | 9.445 |
| Ours | PLCC | 0.873 | 0.896 |
| (Full) | SRCC | 0.845 | 0.893 |
| (weights ✓) | KRCC | 0.652 | 0.703 |
| (ResNet101) | RMSE | 6.728 | 9.220 |
| Ours | PLCC | 0.871 | 0.895 |
| (Full) | SRCC | 0.852 | 0.898 |
| (weights ✓) | KRCC | 0.646 | 0.715 |
| (ViT-base) | RMSE | 6.692 | 9.004 |

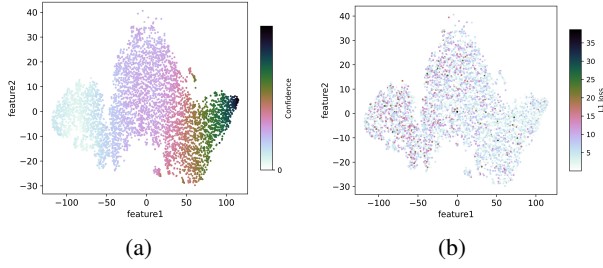

(a)         (b)

*Figure 5.* Comparison of (a). confidence weights of pseudo-labels and (b). $L_1$ loss between pseudo-labels and corresponding GT labels in feature space.

### E.2. Visualization of Label Confidence

In this subsection, we visualize the confidence of pseudo-labels learned by the module of Label Optimizing in Section 3.3 for the 60% unlabeled training images sampled from KonIQ-10K (1:3:1). To be specific, through feature dimensionality reduction by t-SNE, we visualize these 60% unlabeled samples in a two-dimensional space, and make a comparison between the confidence coefficients of their pseudo-labels and the differences from corresponding GT MOS labels (measured by $L_1$ loss).

According to Figure 5, the higher the confidence (left is smaller than right), the smaller the corresponding $L_1$ loss (left is greater than right) on the whole, which proves the validity of Confidence of Labels illustrated in Section 3.3.

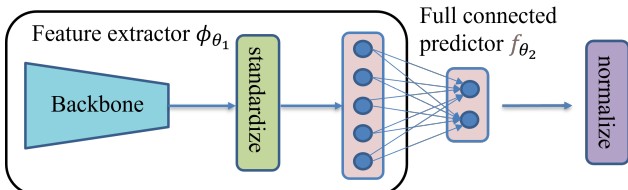

*Figure 6.* The schematic diagram of standardized feature extractor designed for Cosine similarity (CS)-based manifold structure, where the extracted features are standardized.

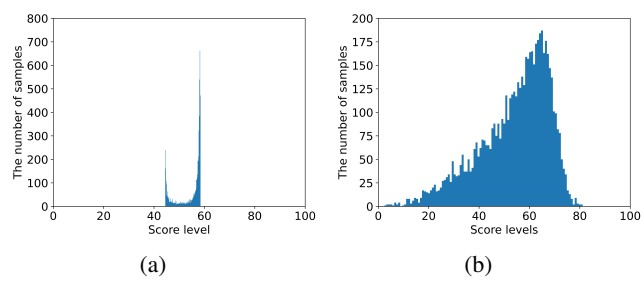

(a)         (b)

*Figure 7.* Comparison of distributions between (a). Pseudo-labels learned based on the CS-based manifold and (b). Ground-truth (GT) MOS labels of the 60% unlabeled training images sampled from KonIQ-10K (1:3:1).

### E.3. Cosine Similarity-based Manifold Structure

In the module of Analysis and Discussion of Section 3.3 in the submitted version, we claimed that the distribution of the predicted MOS labels will be damaged if the sample features are standardized to construct the manifold structure by cosine similarity (abbreviated to CS-based manifold structure for convenience). Here we conduct the following experiments to verify this point.

E.3.1. EXPERIMENT SETTINGS.

In order to make a clear comparison with the results of $k$NN-based manifold structure in Figure 4 in the submitted version, we train our model on KonIQ-10K (1:3:1) with $m = 100$ according to standardized features extracted by ResNet101 and its subsequent FC layer (mapping 2048-dimensional vectors to 256-dimensional features). As shown in Figure 6, different from the feature extractor used in Figure 2 in the submitted version, an operation of feature standardization is embedded in the feature extractor $\phi_{\theta_1}$.

Note that standardization is different from the normalization after predictor $f_{\theta_2}$ in Figure 6. Standardization means standardizing to unit vectors, and normalization means the operation of Softmax. After training the standardized feature extractor, the image quality features can be obtained with norms equal to 1. Therefore, we can define the CS-based

*Table 8.* The impact on CPL-IQA of the number of nearest neighbors $k$ in Eq. 3.

| $k$ | 2 | 4 | 6 | 8 | 9 | 10 | 11 | 12 | 14 |
|------|-------|-------|-------|-------|-------|-------|-------|-------|-------|
| PLCC | 0.812 | 0.847 | 0.838 | 0.859 | 0.865 | 0.873 | 0.870 | 0.864 | 0.868 |
| SRCC | 0.785 | 0.823 | 0.840 | 0.836 | 0.840 | 0.845 | 0.837 | 0.832 | 0.851 |

manifold structure based on the cosine similarity between pairwise sample features:

$$G_{ij} := \begin{cases} q_i^\top q_j & \text{, if } q_i \text{ is linked to } q_j , \\ 0 & \text{, otherwise,} \end{cases} \quad (24)$$

where $q_i$ is linked to $q_j$ means $q_i$ is the $k$ nearest neighbor of $q_j$ in this manifold graph. Everything else stays the same except that Eq. 24 replaces Eq. 2 in the submitted version.

### E.3.2. EXPERIMENT RESULT.

As shown in Figure 7, the distribution of the predicted pseudo-labels becomes high on both sides and low in the middle, which is a far cry from the GT distribution. This supports our earlier mention in the main text that during label propagation (Zhou et al., 2003), features are normalized to construct the manifold structure of samples (Iscen et al., 2019) using cosine similarity, which is not suitable for regression and IQA tasks.

### E.4. The Impact on CPL-IQA of The Number of Nearest Neighbors $k$ in Eq. 3

In Section 3.3.2, the number of nearest neighbors $k$ in Eq. 3 is by default set to 10 in the proposed CPL-IQA. To validate its effectiveness, we conduct experiments with different values of $k$ on the KonIQ-10K dataset. Except for the variation in $k$, the experimental setup remains consistent with that in Table 1. The PLCC and SRCC results with different settings of $k$ are summarized in Table 8.

According to Table 8, it can be observed that when $k$ exceeds 10, there is little to no improvement in PLCC and SRCC. Therefore, we choose to set $k = 10$ as the default value.

### E.5. Performances of CPL-IQA Trained with Labeled and Unlabeled Samples from Different Datasets

In a standard semi-supervised learning framework, the training dataset typically comprises both labeled and unlabeled samples. In our experiments in the main text, we exclusively investigated the scenario where both labeled and unlabeled samples were drawn from the same dataset. To further validate the effectiveness of our proposed semi-supervised CPL-IQA approach, we extended our investigation to a more challenging cross-dataset setting, where labeled and unlabeled training samples are sourced from different datasets. Specifically, we conducted comprehensive experiments to com-

pare the performance of our method against some advanced baselines under this cross-dataset configuration. Specifically, we select 80% from BID (Ciancio et al., 2010), 70% from KonIQ-10k (Hosu et al., 2020), 10% from KonIQ-10k, and the remaining 20% from two datasets as labeled, unlabeled, validation, and test sets, respectively. Among them, the labeled dataset contains 469 images, which is also much smaller than the unlabeled dataset, including 7051 images. The details of dataset BID are shown in Table 9. In this experimental setup, we compared our approach with the following baseline methods: BRISQUE (Mittal et al., 2012a), CORNIA (Ye et al., 2012), NIQE (Mittal et al., 2012b), ILNIQE (Zhang et al., 2015), HOSA (Xu et al., 2016), dipIQ (Ma et al., 2017a), DB-CNN (Zhang et al., 2018), Meta-IQA (Zhu et al., 2020), HyperIQA (Su et al., 2020), UNIQUE (Zhang et al., 2021), and Semi-IQA (Li et al., 2024). The results are shown in Table 10.

From Table 10 we can observe that: **(1)** Comprehensive experimental results demonstrate the superior performance and robustness of our proposed method in cross-dataset scenarios, where labeled and unlabeled training samples originate from different datasets. **(2)** The comparative analysis reveals that CPL-IQA achieves more significant performance improvements on KonIQ-10k compared to Semi-IQA, which substantiates the effectiveness of our proposed label propagation-based semi-supervised IQA framework. The relatively marginal improvement on BID can be attributed to the substantial disparity between the number of labeled and unlabeled samples. Unlike conventional methods that directly predict pseudo-labels using the initial model, CPL-IQA's label propagation mechanism enables more accurate pseudo-label estimation. This characteristic makes CPL-IQA particularly effective in leveraging large-scale unlabeled datasets like KonIQ-10k, where it can fully exploit the abundant unlabeled samples through its propagation-based learning paradigm.

### E.6. Evaluating CPL-IQA Performance with Unlabeled Training Data from Multiple Sources

To systematically investigate the impact of different sources of unlabeled training samples on the performance of the CPL-IQA model, we conducted comparative experiments by adjusting the source of unlabeled training samples while maintaining the experimental setup of Table 1. Specifically, during the training process, we consistently used 20% of the KonIQ-10k dataset as labeled training samples and an

*Table 9.* The detailed attributes of three image databases used in experiments in Appendix E.5 and Appendix E.6. Same to the experiments in the main text, MOS labels are firstly normalized to the MOS range [1, 100] by Max-min Normalization Eq. 1 before training.

| Databases | Number | MOS Range | *Distortion Type* |
|---|---|---|---|
| KonIQ-10K | 10,073 | [1,5] | Authentic |
| BID | 586 | [0,5] | Authentic |
| SPAQ | 11,125 | [0,100] | Authentic |
| KADID-10k | 10,125 | [1,5] | Synthetic |

*Table 10.* Correlation between model predictions and MOSs on the test sets of BID and KonIQ-10k, respectively. Top two correlations are highlighted in boldface.

| Methods | BID | | KonIQ-10k | |
|---|---|---|---|---|
| | PLCC | SRCC | PLCC | SRCC |
| BRISQUE | 0.602 | 0.581 | 0.079 | 0.020 |
| CORNIA | 0.619 | 0.591 | 0.534 | 0.418 |
| NIQE | 0.453 | 0.440 | 0.531 | 0.524 |
| ILNIQE | 0.526 | 0.502 | 0.530 | 0.505 |
| HOSA | 0.561 | 0.501 | 0.652 | 0.633 |
| dipIQ | 0.152 | 0.018 | 0.434 | 0.228 |
| DB-CNN | 0.705 | 0.683 | 0.688 | 0.635 |
| MetaIQA | 0.686 | 0.653 | 0.623 | 0.570 |
| HyperIQA | **0.827** | **0.797** | 0.715 | 0.661 |
| UNIQUE | 0.666 | 0.662 | 0.749 | 0.713 |
| Semi-IQA | 0.811 | **0.801** | **0.775** | **0.750** |
| **Ours** | **0.819** | 0.784 | **0.780** | **0.772** |

additional 20% as test samples. For the unlabeled training samples, we designed three experimental scenarios (with consistent numbers of unlabeled samples across scenarios): (1) Same-source dataset scenario: Unlabeled samples were drawn from the same dataset as labeled samples, utilizing the remaining 60% (6044 samples) of the KonIQ-10k dataset; (2) Same-type distortion dataset scenario: Both labeled and unlabeled samples were from authentic distortion datasets but different sources, with 6044 samples randomly selected from the SPAQ dataset (Fang et al., 2020); (3) Different-type distortion dataset scenario: Unlabeled samples were from a synthetic distortion dataset, with 6044 samples randomly selected from the KADID-10K dataset (Lin et al., 2019).

The detailed characteristics of the SPAQ and KADID-10K datasets are presented in Table 9. The experimental results are documented in Table 11, revealing the following key observations: (1) The CPL-IQA model achieves optimal performance when the unlabeled training set shares the same dataset source with both the test samples and labeled training samples. (2) The model demonstrates superior performance with SPAQ as the unlabeled training set compared to KADID-10K, which can be attributed to the fact that SPAQ, similar to the test samples and labeled training samples,

*Table 11.* PLCC and SRCC Results of CPL-IQA on 20% KonIQ-10k Dataset with Different Unlabeled Training Samples. During the training process, labeled samples were consistently maintained at 20% of the KonIQ-10k dataset across all three experimental settings. The unlabeled training samples were respectively derived from: (1) 60% of KonIQ-10k (6044 samples), (2) an equivalent number of samples from SPAQ dataset, and (3) an equivalent number of samples from KADID-10K dataset.

| Unlabeled Training Sets | KonIQ-10k | SPAQ | KADID-10k |
|---|---|---|---|
| PLCC | 0.873 | 0.844 | 0.832 |
| SRCC | 0.845 | 0.833 | 0.817 |

belongs to the category of authentically distorted datasets. These findings collectively suggest that the model's performance on the test set (from the same dataset as the labeled training samples) improves when the unlabeled training set exhibits closer similarity to the labeled training samples.

