# OpenReview forum: "Semi-Supervised Blind Quality Assessment with Confidence-quantifiable Pseudo-label Learning for Authentic Images"
_ICML.cc/2025/Conference — ICML 2025 poster_

### Official Review · Reviewer_cfTH · 2025-02-17

**Overall Recommendation:** 4

**Summary:**

This work aims at addressing the quality assessment task for authentically disorted images from a semi-supervised learning perspective. The key idea is to leverage confidence quantifiable pseudo-label learning to confront the data insufficiency challenge. The proposed CPL-IQA can be trained on unlabeled images with an entropy-based confidence learning to mitigate the negative effects from noisy labels.


## update after rebuttal
In the rebuttal, the authors have futher verified the value of this work by leveraging unlabeled data for boost IQA model. I will keep my rating.

**Claims And Evidence:**

Yes,

**Essential References Not Discussed:**

No.

**Experimental Designs Or Analyses:**

In the experiments, existing IQA datasets with full quality annotations are manually splited into three parts: training images with labels, training images without labels (exists but not used), and test images. The experiments setup is reasonable to verify the effectiveness of the proposed semi-superivsed learning scheme. Compared with supervised learning, unsupervised learning, and other semi-supervised learning methods in the filed of IQA, the proposed method presents better performance. To the Reviewer, this work can be further improved if the authors can use the proposed method to train an IQA model on the combination of existing annotated IQA datasets and extra unlabeled images following regular setup, e.g., split KonIQ-10k with a ratio of 8:2 (training/test), training the model on the training set of KonIQ-10k and an extra unlabeled dataset, verifying that the the performance of model is improved on the test set of KonIQ-10k compared to the counterpart which is trained only on training set of KonIQ-10k.

**Methods And Evaluation Criteria:**

Yes. This work propose to leverage unlabeled data for IQA model training, which includes an neccessary module to mitigate the inevitable label noise issue. The proposed alternate learning between model training and label optimization is reasonable since the reliability of pseudo-labels progressively increase with the correctness of the model predictions.

**Other Comments Or Suggestions:**

No.

**Other Strengths And Weaknesses:**

Resizing all images into 256x256 may deteriorate IQA performance, it's better to crop multiple image patches from images with their original resolution.

**Questions For Authors:**

In Table 10, the SRCC result of HyperIQA is higher than the proposed method but not marked with bold. Is there any typo?

**Relation To Broader Scientific Literature:**

In the era of deep learning, data plays a key role in developing powerful artificial intelligence models across various tasks. However, the scale of data will be limited when the cost of annotations is high, where IQA is one of such case. This work is closely related to the important topic of training artificial intelligence models using unlabeled data.

**Theoretical Claims:**

The proofs in the appendix is correct to the Reviewer.

---

> ### Author Rebuttal · Authors · 2025-03-31
>
> Thank you for your affirmation and the valuable comments that help us improve our work. The following is a careful response and explanation about the weaknesses and questions.
>
> # For Experimental Designs Or Analyses
> We sincerely appreciate your recognition of our experimental design and analysis, as well as this valuable suggestion.
>
> In fact, we have already applied the proposed method to train an IQA model using a combination of existing annotated IQA datasets and additional unlabeled images under standard experimental settings. Specifically, as shown in Appendix Table 10, we select 80% from BID,70% from KonIQ-10k, 10% from KonIQ10k,and the remaining 20% from two datasets as labeled, unlabeled, validation, and testsets, respectively. The results clearly demonstrate the superiority of our approach.
>
> To further validate the effectiveness of our method, as suggested by the reviewer, we split KonIQ-10k into labeled training and test sets (8:2 ratio), trained the model on the labeled KonIQ-10k training set and the unlabeled LIVE-C dataset, and evaluated it on the remaining 20% KonIQ-10k test set. The results are as follows:
>
> | training sets | 80% labeled KonIQ-10k | 80% labeled KonIQ-10k + unlabeled LIVE-C |
> |-|-|-|
> | PLCC | 0.879 | **0.891**  |
> | SRCC | 0.874 | **0.885** |
>
> From the above results, we can observe that, after training with the extra unlabeled dataset LIVE-C, the performance of the model is improved on the test set of KonIQ-10k compared to the counterpart which is trained only on training set of KonIQ-10k. These results further validate the effectiveness of our proposed method.
>
> In the camera-ready version, we will incorporate these additional experimental findings to further strengthen our work.
>
> # For Other Strengths And Weaknesses
> Thanks for this valuable comment. We acknowledge that resizing could theoretically alter distortion characteristics and affect quality assessment accuracy. In fact, the resize-and-crop strategy in our model is actually another choice that is widely used in many works, which can balance computational efficiency and practical applicability. Below, we provide a detailed justification for this design choice in our model from three perspectives:
> 1. **Preservation of Global Context:** Since the required final cropped size during training is 224×224, resizing images to 256×256 before cropping in our method ensures that the model captures relatively global regions of the image while retaining fine-grained distortion patterns. This strategy has been widely adopted in classic IQA works such as DB-CNN, NIMA, and SSL-IQA, and our approach follows this well-established practice.
> 2. **Trade-off Between Performance and Efficiency:** We agree with the reviewer that cropping multiple patches from original-resolution images (without resizing) is theoretically preferable. However, this requires extracting multiple patches per image, significantly increasing training overhead and computational complexity. This is because, for large original images, a small number of cropped patches may fail to cover sufficient global regions of the image. Therefore, existing models adopting this strategy (e.g., LIQE and HyperIQA) typically address this by cropping a large number of patches, which further escalates computational costs.
> 3. **Empirical Validation:** Following the experimental setup in Table 1, we conducted additional tests by randomly cropping 10 patches per original-resolution image (denoted as Strategy 2) and compared its performance with our default resize-and-crop approach (Strategy 1). The results show that Strategy 2 only improves PLCC and SRCC by less than 1%, but at the cost of 4× longer runtime. The cost-performance ratio is unfavorable.
>
> | Pre-processing methods | Stategy 1 | Stategy 2 |
> |-|-|-|
> | PLCC | 0.873 | 0.878  |
> | SRCC | 0.845 | 0.851 |
> | Running time in one epoch of Stage 1 (s) | 63.24 | 238.57 |
>
> We will emphasize in camera-ready version that our resizing-and-cropping strategy balances efficiency and performance, ensuring scalability for real-world deployment.
>
> # For Questions
> Thank you for your careful review and valuable feedback. **We are sorry for this oversight and will correct it in the camera-ready version** by bolding the SRCC result of HyperIQA in the table.
>
> Notably, this oversight does not impact Appendix E.5's analysis, as Table 10 primarily compares our method with Semi-IQA, where ours shows superior performance. While HyperIQA achieves marginally better results on BID through computationally intensive supervised training, it underperforms significantly on KonIQ-10k compared to our method, further validating our approach's effectiveness.
>
> In camera-ready version, we will conduct a more thorough review to eliminate any remaining typos and ensure the rigor and high standard of our research.

---

### Official Review · Reviewer_s6Ro · 2025-03-11

**Overall Recommendation:** 4

**Summary:**

This paper presents a novel semi-supervised blind image quality assessment framework, named CPL-IQA, for assessing the quality of real distorted images. The method effectively utilizes a large number of unlabeled real distorted images through confidence-quantifiable pseudo-label learning, addressing the challenge of limited labeled data in the field of image quality assessment.CPL-IQA consists of two phases, namely, label transformation preprocessing and alternating model training and label optimization. Its core innovation lies in the label optimization strategy based on the streaming assumption and the confidence learning method for pseudo-labels. Experimental results show that the framework performs well on real distorted image datasets, provides a more standard semi-supervised learning paradigm, and does not require additional supervisory information or complex network structure.

## update after rebuttal
The author's response answered my questions and I will keep my score. In addition, I hope the author will finally make the code public and promote the community.

**Claims And Evidence:**

1) The main innovations of CPL-IQA include a label optimization strategy based on the streaming assumption and a confidence learning method for pseudo-labeling that enhances reliability and mitigates the effects of outliers. Section 3.3.3 of the paper describes the label optimization process in detail, and Section 3.3.4 describes the confidence learning method and demonstrates the effectiveness of these methods in the experimental section through ablation studies and visual analysis.
2) Experimental results show that the framework exhibits superior performance on real-world distorted image datasets. Table 1 shows the performance comparison of CPL-IQA with other traditional and deep learning BIQA methods on the KonIQ-10K dataset, and the results show that CPL-IQA achieves better performance metrics. The cross-dataset experiments in Table 2 also further validate the generalization ability of CPL-IQA.
3) The proposed confidence learning approach eliminates pseudo-label outliers and enhances the generalization capability of CPL-IQA. Section 3.3.4 describes the principle of confidence learning and the relationship between confidence and pseudo-label accuracy is supported by the visual analysis in Figure 5. The ablation study in Table 3 also shows the performance enhancement of using confidence weights.

**Essential References Not Discussed:**

The relevant work cited in the paper is relatively comprehensive and does not suffer from missing key literature.

**Experimental Designs Or Analyses:**

- CPL-IQA was compared with sixteen state-of-the-art BIQA methods, both traditional and deep learning-based, on the KonIQ-10K dataset. The experimental setup uses a 1:3:1 dataset division ratio (training labeled data: training unlabeled data: test data). The results are presented in Table 1.

- The effects of the different components of CPL-IQA were analyzed experimentally. Specifically, the effects of Label Conversion, the weight of Label Confidence, and Label Optimization in Stage 2 are investigated, and the results are presented in Table 3. In addition, the effects of the base m of the score set M and the proportion of dataset partitioning on the performance are also investigated, and the results are presented in Tables 4 and 5, respectively.

- The quality of the pseudo-labels learned by Eq. (7) in each iteration of Stage 2 was analyzed by visual means. The change in performance between pseudo-labeling and model-directly predicted labeling was compared (Fig. 3), as well as the pseudo-labeling distribution versus the true MOS labeling distribution (Fig. 4).

**Methods And Evaluation Criteria:**

1) The paper clearly states that the problem to be addressed is semi-supervised blind image quality assessment (BIQA) in real distortion scenarios since it is difficult to obtain distortion-free reference images in practical applications and the cost of labeling real distortion images is high. The CPL-IQA framework is precisely designed to utilize limited labeled data and a large number of unlabeled real distorted images. The core idea is to enhance the performance of the model in the data sparse case through confidence quantized pseudo-label learning.
2) The paper uses a series of representative real distortion image databases as benchmark datasets, including KonIQ-10K, LIVE-C, NNID, and SPAQ. These datasets are commonly used in the field of BIQA, and in particular, KonIQ-10K and SPAQ are considered to be the most important datasets containing real-world distortions. The use of these datasets can effectively evaluate the performance of models in real-world application scenarios.

**Other Comments Or Suggestions:**

I hope that in the future the authors will make the code public and promote the community.

**Other Strengths And Weaknesses:**

Strenghts:

- The originality lies in its novel semi-supervised BIQA framework CPL-IQA.
Originality lies in its novel semi-supervised BIQA framework, CPL-IQA, which efficiently utilizes unlabeled real distorted images through confidence-quantifiable pseudo-label learning, which directly addresses the key challenge of scarcity of labeled data in the BIQA domain.
- Uniqueness of Label Transformation Strategy: The entropy minimization-based label transformation method proposed by CPL-IQA is a key innovation in transforming scalar MOS labels into vector labels. This paves the way for applying label propagation techniques, which excel in handling vector labels, to regression tasks like MOS prediction, and draws on the idea of utilizing MOS distributions for training in NIMA, but with innovative adaptations.
- Successful application of label propagation to regression tasks: traditional label propagation methods are mainly applied to classification problems. cPL-IQA successfully extends the idea of label propagation to regression tasks (i.e., image quality assessment) and proposes a corresponding implementation via Eq. (7), which is in itself of some originality and importance.

Weaknesses:

- Although Appendix C provides the algorithm pseudo-code, the key steps of the training process, such as label transformation, graph construction, label optimization, and the iterative process of model training, can be described in more detail in the main method section, which helps readers better understand the implementation details of the algorithm.

-  While the paper provides proofs of label propagation convergence and equivalence to the regularization framework (in the Appendix), a more concise exposition of the intuitive understanding of these theoretical linkages in the main body might help a broader audience understand the theoretical underpinnings of the approach. For example, it could be briefly explained how the flow shape assumption plays a role in label propagation and how the regularization framework explains the process of label propagation from an optimization perspective.

**Questions For Authors:**

The entropy minimization-based label transformation method proposed in the paper is a key step in transforming scalar MOS labels into vector labels. Although it is explained in the paper that this is done to better model the MOS distribution and obtain higher confidence, is this conversion strategy still robust in the face of different datasets or large differences in the characteristics of the MOS distribution? Have the authors conducted relevant experiments to verify the generalization ability of this label transformation strategy, e.g., analyzing the quality of the transformed vector labels on datasets with significantly different MOS distributions? I would consider the method more practical if the authors could provide an analysis of the label transformation effect across datasets or comparative experiments to demonstrate the robustness of the strategy. If the label transformation is very sensitive to the data characteristics, it will limit the scope of its application in different scenarios in the real world.

**Relation To Broader Scientific Literature:**

- Semi-supervised Blind Image Quality Assessment (BIQA) Framework: The core of the CPL-IQA proposal lies in its semi-supervised learning paradigm, which aims to address the problem of scarcity of labeled data in real-world scenarios. This is in line with the recent trend of research on semi-supervised and unsupervised BIQA methods. Existing deep learning (DL)-driven BIQA methods are usually “data starved”, and thus utilizing unlabeled data to improve performance has become an important research direction.CPL-IQA efficiently utilizes unlabeled real distorted images through pseudo-label learning with quantifiable confidence. This is in line with some previous semi-supervised BIQA works such as SSLIQA (Yue et al., 2022) and SS-IQA (Pan et al., 2024), which all aim to utilize unlabeled data, but CPL-IQA differs in that it uses only a single-branch network and does not rely on additional inputs or datasets, which reduces the training cost and improves applicability, whereas SSLIQA and SS-IQA are more suitable to utilize unlabeled data, whereas CPL-IQA and SS-IQA are more suitable to utilize unlabeled data. SSLIQA and SS-IQA require multi-branch networks and additional data.

- Confidence learning method for pseudo-labeling: In order to mitigate the negative impact of inaccurate pseudo-labeling on model training, CPL-IQA proposes an entropy-based confidence learning method. The uncertainty of the predicted pseudo-labels is estimated by calculating their entropy; the higher the entropy, the lower the confidence level, and a lower weight will be assigned in subsequent model training. This is consistent with the common strategy of handling noisy pseudo-labels in semi-supervised learning.

**Theoretical Claims:**

- Assertion 3.1 claims that the sequence ${P^{(t)}}$ defined by Eq. (5) converges to $P^*$ in Eq. (6) via $\tilde{G}$ obtained by Eq. (4). The proof of this statement is given in Appendix A.1.

- Assertion 3.2 claims that the limit value of the iterative process of Eq. (5), Eq. (6), can be considered as an optimal solution of the regularization framework Eq. (13) (with $\mu > 0$ in Eq. (14)). The proof of this statement is given in Appendix A.2.

---

> ### Author Rebuttal · Authors · 2025-03-31
>
> Thanks for your affirmation and the valuable comments that help us improve our work. The following is a careful response and explanation about the weaknesses and questions.
>
> # For Weakness 1
> We sincerely appreciate the constructive feedback. We would like to clarify that Sections 3.3.1 to 3.3.6 (pages 4–6) of the manuscript have already provided a comprehensive and step-by-step explanation of the algorithm’s implementation.
>
> To further enhance clarity and ensure readers can better understand the implementation details, **in the final version**, we will include more illustrative examples or pseudocode snippets within the main text to complement the existing Algorithm 1 in Appendix C, which can further improve the readability and transparency of the method while maintaining the rigor of the technical content.
>
> # For Weakness 2
> Thank you for the insightful suggestion. **In the final version, we will add a concise discussion in Section 3.3.7 for a more concise exposition of the intuitive understanding of these theoretical linkages**, including the following two key insights:
>
> **On the one hand**, the manifold assumption serves as the theoretical foundation for label propagation, positing that samples with similar features (i.e., neighboring nodes in the graph) should share similar labels. This fundamental assumption enables the label propagation process, where labels from annotated samples are effectively disseminated to unlabeled ones through the constructed nearest neighbor graph (Eq.4), thereby facilitating pseudo-label learning.
> **On the other hand**, for the optimization perspective, the iterative propagation (Eq. 5) implicitly solves the regularization problem (Eqs. 13-14), where the first term enforces smoothness (labels vary slowly over the constructed nearest neighbor graph), and the second term fits the initial labels. The hyperparameter $\mu$ in Eq.14 balances the above two objectives.
>
> # For Questions
> Thank you for the valuable comments that help us improve our work. In fact, **both of the proposed model and label conversion strategy exhibit strong robustness across diverse datasets with varying MOS distributions**, which have been comprehensively demonstrated in our paper. Specifically:
>
> **On the one hand**, we have conducted experiments on multiple datasets (KonIQ-10K, SPAQ, LIVE-C, and NNID) with significantly different MOS ranges and distribution characteristics (e.g., KonIQ-10K: [1,5], SPAQ: [0,100]). The consistent performance improvement (Tables 1–3) across these datasets demonstrates the generalization ability of our label conversion strategy.
>
> Additionally, in Appendix E.5 and E.6, we explicitly tested scenarios where unlabeled training data came from different distortion types (authentic vs. synthetic) and sources (e.g., training with labeled dataset BID and unlabeled datasets KonIQ-10K in Appendix E.5, and training with labeled dataset KonIQ-10K and unlabeled datasets SPAQ/KADID-10K in Appendix E.6). Results in Tables 10 & 11 show that our method remains effective even when MOS distributions differ, as the entropy minimization inherently adapts to the input label range (Eq. 1 normalizes MOS to [1,100]) and preserves relative quality relationships. The confidence-weighted training (Eq. 8–9) further mitigates potential outliers from label conversion.
>
> To further validate the robustness of our proposed strategy, we conducted additional experiments: we split KonIQ-10k into labeled training and test sets (8:2 ratio), trained the model on the labeled KonIQ-10k training set and the unlabeled LIVE-C dataset, and evaluated it on the remaining 20% KonIQ-10k test set. The results are as follows:
>
> | training sets | 80% labeled KonIQ-10k | 80% labeled KonIQ-10k + unlabeled LIVE-C |
> |-|-|-|
> | PLCC | 0.879 | **0.891**  |
> | SRCC | 0.874 | **0.885** |
>
> The above results show that our method is robust to different datasets.
>
> **On the other hand**,  **Although the label conversion process in Eq. (2) cannot be directly applied to predict MOS distributions** (since our method operates under the reasonable assumption that the labeled samples have sufficiently high confidence levels (Lines 190-192)), **the well-trained CPL-IQA model can robustly predict label distributions.** For example, using the model trained as described above, we randomly selected 10 samples each from KonIQ-10k and LIVE-C, and computed the average JS divergence and Wasserstein distance between the predicted MOS distributions and the variance-simulated normal distributions (approximating ground truth). The results are as follows:
>
> | Samples | KonIQ-10k |  LIVE-C |
> |-|-|-|
> | JS | 0.065 | 0.097  |
> | W-Dist | 0.042 | 0.086 |
>
> The results demonstrate that both the JS divergence and Wasserstein distance remain below 0.1, confirming the accuracy of the predicted MOS distributions and validating the robustness of our method across diverse datasets and MOS distributions.
>
> **In final revision, we will further highlight the robustness.**

---

> > ### Comment · Reviewer_s6Ro · 2025-04-06
> >
> > I hope the author will finally make the code public and promote the community. The author's response answered my questions and I will keep my score.

---

> > > ### Author Response · Authors · 2025-04-06
> > >
> > > Thank you for your recognition and support of our work! We will make the code publicly available after the paper is accepted.

---

### Official Review · Reviewer_mdhL · 2025-03-12

**Overall Recommendation:** 2

**Summary:**

This work focuses on semi-supervised blind image quality assessment (BIQA). The method first converts MOS labels to vector labels via entropy minimization, then constructs nearest neighbor graph to help label optimization with confidence. The pseudo labels are then combined with ground truth to guide the model retraining. The experiments show a good performance of the proposed method.

------------------------- ## update after rebuttal ##-------------------------------------

comments after rebuttal:

The authors have partially addressed my concerns, but some confusions still exist.

(1) The performance of different conversion methods given in the rebuttal  shows an evident fluctuation, which is somewhat weird. From a natural sense, the performance should not be that sensitive, especially for the N-D simulation with fixed variance, which is approximately a smoothed version of the proposed entropy maximization.

(2) The authors claims that "our approach maintains compatibility with any single-branch backbone architecture" in the rebuttal, which is not supported by current version of experiments.

Overall, I would keep my score unchanged.

**Claims And Evidence:**

The authors claim that  images with similar MOS values may correspond to a variety of score distributions (it is true) as shown in Fig.1, but in the label conversion, the variation is not considered, which looks like paradoxical.

**Essential References Not Discussed:**

The references are almost adequate in a constrained page space.

**Experimental Designs Or Analyses:**

Almost sound but inadequate. It would be better to give more experiments. For example, how does the method perform if B_L and B_u varies. Further, the method requires at least two-step training, which puts more burden on the implementation, and the training cost is also questionable.

**Methods And Evaluation Criteria:**

It does make sense.

**Other Comments Or Suggestions:**

I have put all my comments and questions in the above.

**Other Strengths And Weaknesses:**

[strength]
1) The work designs a semi-supervised BIQA method to address the limited labeled data.
2) The method introduces a nearest neighbor graph and label propagation strategy to construct pseudo labels.
3) The method is sound and easy to understand.

[weakness]
1) The label conversion is somewhat common in IQA/VQA. For example, Q-Align adopts a similar conversion method. It is also good to compare the proposed label conversion with existing MOS discretization methods to demonstrate the distinction and effectiveness.
2) The experimental result is not impressive comparing to current methods such as SSLIQA.
3) It would be better to give more experiments. For example, how does the method perform if B_L and B_u varies. I'm wondering if the performance is sensitive to the ratio of B_u/B_L, and which value is more reasonable.
4) The method requires at least two-step training, which puts more burden on the implementation, and the training cost is also questionable.

**Questions For Authors:**

I have put all my comments and questions in the above.

**Relation To Broader Scientific Literature:**

Increamental. The label conversion is not uncommon, which has been tried in various work in quality assessment.

**Theoretical Claims:**

Theoretically correct.

---

> ### Author Rebuttal · Authors · 2025-03-31
>
> We appreciate the valuable comments and will improve in final version.
> # For Claims & Evidence
> This is a misunderstanding. We clarify that:
>
> 1. **The purpose of Fig. 1 is to demonstrate the limitations of some existing semi-supervised BIQA methods that require full score distributions (Left Lines 61-68)**. As shown, images with similar MOS values can have divergent score distributions, making direct MOS-to-distribution conversion unreliable. Unfortunately, many datasets lack distribution information, limiting these semi-supervised methods' applicability.
>
> 2. While some datasets provide variance information, simulating a normal distribution solely based on variance is often inaccurate. Moreover, in real-world scenarios, variance information is frequently unavailable. These limitations restrict the broad applicability of these semi-supervised methods.
> In addition, as shown in Left Line 95, it is confirmed that **models trained with vectorized distribution labels outperform those using scalar MOS labels**.
> Therefore, the two conflict findings motivated us to **develop a more flexible solution that converts MOS labels into vector labels for training without requiring distribution and variance**.
>
> 3. Therefore, **the vectorized conversion process in our method does not depend on variance information**. To achieve this, we propse the entropy minimization (Eq.2) under the **high-confidence MOS assumption** (Left Lines 182-201), leading both better performance than scalar MOS training and broader applicability than distribution-dependent methods.
>
> Thus, Fig. 1 and our method are logically consistent. **We will clarify this point more explicitly in final version to prevent any potential misunderstanding.**
>
> # For Weakness 1
> We acknowledge that label conversion is relatively common in IQA. However, **existing methods are designed specifically for either fully supervised or LLM-based BIQA**, including:
> 1. [1] discretizes the score range into five equal intervals, converting continuous scores into one-hot encoded rating levels.
> 2. [2] converts MOS labels into vector representations by assuming each MOS score follows a normal distribution, which is simulated using a predetermined fixed variance.
> 3. [3] extends [2] by incorporating sample-specific variance values.
>
> We conduct experiments comparing our method with: (1) the one-hot conversion in [1], and (2) the normal distribution (N-D) simulation used in [2-3], using the same setting as Tab. 1.
> | method | One-hot | N-D | Ours |
> |-|-|-|-|
> | PLCC | 0.803 | 0.832  | **0.873** |
> | SRCC | 0.796 | 0.816 | **0.845** |
>
> **Evidently, existing methods are unsuitable for semi-supervised BIQA scenarios due tofailing reliable pseudo-label confidence assessment**. In contrast, our method innovatively addresses this gap.
>
> [1] Q-align. ICML'24
>
> [2] BIQA with Probabilistic Representation. ICIP'18
>
> [3] A fuzzy neural network for opinion score distribution prediction for BIQA. TCSVT'24
>
> # For Weakness 2
> While our method shows marginal improvement over SSLIQA in Tab 1, it demonstrates significantly superior performance in cross-dataset experiments on LIVE-C in Table 2, highlighting its exceptional generalization capability. Moreover, as detailed in Line 715, SSLIQA requires multi-branch network training, whereas our approach maintains compatibility with any single-branch backbone architecture, offering reduced parameters and simpler structure. These advantages substantiate the significance of our method.
>
> # For Weakness 3
> Thanks for your suggestion. In fact, the setting $B_L=8, B_U=56$ in Stage 2 is to match Stage 1's batch size $64$, i.e., $B_L+B_U=64$.
> Moreover, the significantly larger size of $B_U$ compared to $B_L$ reflects the practical scenario where labeled samples are substantially fewer than unlabeled samples.
>
> To investigate the impact of ratio $B_L / B_U$, under the settings of Tab. 1, we obtained the following results:
>
> | $(B_L,B_U)$ | (4,60) | (8,56) | (16,48) |
> |-|-|-|-|
> | PLCC | 0.869 | 0.873  | 0.871 |
> | SRCC | 0.841 | 0.845 | 0.848 |
>
> The results show that while an excessively small
> $B_L$ may degrade performance; **the model exhibits relative insensitivity** to variations in the ratio of $B_u / B_L$.
>
> # For Weakness 4
> This is a misunderstanding. The two-step training in our method **does not put a burden on the implementation**.
>
> As stated in Sec. 3.2, our framework employs a two-stage approach:
> **Stage 1** utilizes only labeled data for training, introducing no additional computational overhead;
> **Stage 2** consists of two steps, with the pseudo-labeling process requiring merely 2.1% of the total training time **(see Response to Reviewer R7vR, Question 2)**.
>
> Moreover, training on SPAQ, our method requires only 16.69 minutes (1 epoch in Stage 1 + 1 epoch in Stage 2) on a single 2080Ti GPU, versus SSLIQA's 18.31 minutes for 2 epochs. This efficiency gain stems from our single-branch design versus SSLIQA's dual-branch architecture, demonstrating our superior efficiency.

---

### Official Review · Reviewer_R7vR · 2025-03-14

**Overall Recommendation:** 3

**Summary:**

In this paper, an algorithm named CPL-IQA is proposed for semi-supervised BIQA task. The proposed algorithm leverages confidence-quantifiable pseudo label learning to utilize the unlabeled images for training. Specifically, it first converts MOS labels to vector labels via entropy minimization. Then, during training, it predicts the pseudo label of unlabeled images by NN graph construction. CPL-IQA alternates model training and label optimization. Extensive experimental results show that the proposed algorithm achieves the better performances than existing algorithms.

**Claims And Evidence:**

Yes, the claims are supported by extensive experimental results including ablation studies and proofs in appendix.

**Essential References Not Discussed:**

Some recent papers are not addressed and compared. It would be better to compare with these algorithms as well.

- [1] Q-Align: Teaching LMMs for Visual Scoring via Discrete Text-Defined Levels, ICML24
- [2] Boosting Image Quality Assessment through Efficient Transformer Adaptation with Local Feature Enhancement, CVPR24
- [3] Blind image quality assessment based on geometric order learning. CVPR24

**Experimental Designs Or Analyses:**

Yes, experimental designs and analysis process are valid and fair.

**Methods And Evaluation Criteria:**

Yes, the proposed algorithm technically sounds and the evaluation process seems fair.

**Other Comments Or Suggestions:**

N/A

**Other Strengths And Weaknesses:**

Please see questions for authors section for weakness.

**Questions For Authors:**

- Figure 4 compares the distribution of pseudo labels and that of corresponding GT labels. However, it would be good to have quantitative measurement such as MAE between pseudo and GT labels.

- Does the pseudo labeling process increase the training time significantly? How fast it is?

- I am curious whether adding more unlabeled data would improve performance. For example, if the KonIQ dataset with labels and the LIVE-C dataset without labels are trained together, how does the performance change when tested on the KonIQ dataset? Table 11 provides similar experimental results. However, it only uses 20% of KonIQ dataset as labeled samples.

- What is the limitation of the proposed algorithm? For example, it would be good to have failure cases.

**Relation To Broader Scientific Literature:**

Most of algorithms for BIQA have been proposed for supervised learning scenario. However, the proposed algorithm is proposed for semi-supervised IQA to perform better on real-world distorted images, which are not well handled in existing IQA datasets for supervised learning scenarios. To this end, the proposed algorithm exploits pseudo label learning approach. This deep semi-supervised learning mechanism is well-studied but not common in BIQA field.

**Theoretical Claims:**

Yes, theoretical claims seem to be correct for me.

---

> ### Author Rebuttal · Authors · 2025-03-31
>
> Thanks for the affirmation and valuable suggestions. We have addressed each point below in detail.
> # For Essential References Not Discussed
> In fact, our method is not directly comparable with [1], [2], or [3], as they follow fundamentally different paradigms. Specifically:
> * **Ours: A semi-supervised BIQA** method that leverages confidence-quantifiable pseudo-label learning to utilize unlabeled data.
> * **[1]: A fully supervised LLM-based** approach trained on multiple joint datasets, requiring massive computational resources.
> * **[2]: A fully supervised** method that adapts **large-scale pretrained models** with minimal trainable parameters.
> * **[3]: A fully supervised** approach using **multiple comparison transformers** and score pivots to construct an embedding space.
>
> Thus, our method is entirely incomparable with [1] and, strictly speaking, also with [2] and [3]. Nevertheless, for a fair comparison under as equal training conditions and model sizes as possible, we conducted experiments following the settings in Table 2 of our paper:
> |  | KonIQ-10K | KonIQ-10K |LIVE-C | LIVE-C | NNID | NNID |
> |-|-|-|-|-|-|-|
> |  | PLCC | SRCC | PLCC | SRCC | PLCC |SRCC |
> |  [2] | 0.868 | 0.836  |0.742 | 0.685 |0.760 | 0.737 |
> | [3] | 0.852 | 0.831 | 0.759 |0.703 |0.746 | 0.758 |
> | **Ours**   | **0.873** | **0.845** | **0.777** | **0.721** | **0.772** | **0.773** |
>
> The results confirm our method's superiority. **In final version, we will cite [1], [2], [3] and expand** the experimental section to include more detailed discussions and comparisons with them, further validating the effectiveness of our approach.
> # For Question 1
> In fact, as mentioned in Left Lines 423-427, Figure 4 primarily serves to demonstrate that the distribution of predicted pseudo-labels closely aligns with that of the ground truth, thereby validating the effectiveness of our Label Optimizing module.
>
> For quantitative evaluation, we have computed the two metrics between pseudo-labels and GT labels, including MAE (5.124) and RMSE (6.687). We will incorporate these results into final version for quantitative analysis.
>
> # For Question 2
> The pseudo-labeling process does not significantly increase training time, primarily due to two key innovations:
> * While the label propagation is an iterative process (Eq. 5), we prove in Assertion 3.1 and Eq. 6 that its convergent solution can be obtained through simple matrix operations (Eqs. 6-7), eliminating the need for actual iterations.
> * For constructing the nearest neighbor graph matrix (Sec 3.3.2), we employ FAISS (as stated in Right Line 303) - an efficient dense vector similarity search library developed by Facebook AI Research, which dramatically accelerates the pseudo-label learning process.
>
> To further validate the efficiency of pseudo-label learning, we tested on a single 2080 Ti GPU using SPAQ with the same experimental setup as in Table 3 (SPAQ 1:8:1). The results show that: (1) Stage 1 training required 63.24 seconds per epoch (5 epochs in total); (2) In Stage 2, pseudo-label generation took 19.38 seconds while model training required 918.56 seconds per epoch (5 epochs in total). This demonstrates that the pseudo-labeling process accounts for only 2.1% of Stage 2 training time and 1.9% of the combined runtime for both stages.
>
> # For Question 3
> Indeed, incorporating additional unlabeled data enhances model performance, and the same holds true for labeled data. While the results in Tables 5 & 10 partially support this observation, we conducted the following supplementary experiments to further validate the conclusion:
>
> **On the one hand**, to systematically validate this effect on KonIQ-10k, we varied the proportion of unlabeled data ($N_u$) following the settings in Table 1:
>
> | $N_u$ | PLCC | SRCC |
> |-|-|-|
> |  0% | 0.850 | 0.822  |
> | 30% | 0.865 | 0.831 |
> | 60%   | **0.873** | **0.845** |
>
> **On the other hand**,  following Table 11's settings, we only increased the labeled training data proportion ($N_l$) from KonIQ-10k while using SPAQ as unlabeled training data:
>
> | $N_l$ | 20% | 50% | 80% |
> |-|-|-|-|
> | PLCC | 0.844 | 0.870  | **0.892** |
> | SRCC | 0.833 | 0.858 | **0.883** |
>
> The above results and Tables 5 & 10 can confirm that expanding labeled or unlabeled datasets can both improve model performance.
> # For Question 4
> The key limitation of our algorithm stems from the label conversion module based on entropy minimization, which inherently presumes high confidence in the annotated quality score distributions. In practice, annotation biases (such as significant labeling noise or uneven MOS distributions) may compromise pseudo-label reliability. For example, when adding Gaussian noise (μ=0.5, σ=0.5) to the MOS labels of training samples while keeping the Table 1 configuration, PLCC and SRCC decreased to 0.632 and 0.607 respectively.
>
> Therefore, our future work will focus on developing noise-robust BIQA techniques. We will further add and analyze these limitations in **final version**.

---

> > ### Comment · Reviewer_R7vR · 2025-04-08
> >
> > Thank you for the detailed response. I will keep my score.

---

> > > ### Author Response · Authors · 2025-04-08
> > >
> > > Thank you for your review. We sincerely appreciate your acknowledgement and recognition of our efforts.

---

### Decision · Program_Chairs · 2025-05-01

**Decision:**

Accept (poster)

**Comment:**

This paper proposes CPL-IQA, a semi-supervised BIQA framework that uses confidence-aware pseudo-labels and entropy-based label conversion to leverage unlabeled data. It achieves strong results on real-world distorted images without added supervision or complexity. Three out of four reviewers are positive. The remaining reviewer raised a few unresolved concerns, stating that (1) the performance of different label conversion methods fluctuates unexpectedly, particularly for the N-D simulation, and (2) the claimed compatibility with single-branch architectures is not fully supported by experimental evidence. Nonetheless, the AC believes the strengths of the paper outweigh these issues and recommends acceptance. The authors are expected to address the above concerns and include the rebuttal content in the camera-ready version and release the code as promised.